# 3D printing of self-healing longevous multi-sensory e-skin
Antonia Georgopoulou[1,2], Sudong Lee[3], Benhui Dai[3], Francesca Bono[1], Josie Hughes[3] & Esther Amstad[1,2] ✉

Electrically conductive hydrogels can simulate the sensory capabilities of natural skin, such that they are well-suited for electronic skin. Unfortunately, currently available electronic skin cannot detect multiple stimuli in a selective manner. Inspired by the deep eutectic solvent chemistry of the frog Lithobates Sylvaticus, we introduce a double network granular organogel capable of simultaneously detecting mechanical deformation, structural damage, changes in ambient temperature, and humidity. The deep eutectic solvent chemistry adds an additional benefit: Thanks to strong hydrogen bonding, our sensor can recover 97% of the Young's modulus after being damaged. The sensing performance and self-healing capacity are maintained within a temperature range of −20 °C to 50 °C for at least 2 weeks. We exploit the granular nature of this system to direct ink to write a cm-sized frog and e-skin wearables. We realize selective tactile perception by training recurrent neural networks to achieve sensory stimulus classification between the temperature and strain with 98% accuracy.

Human skin can detect and classify multiple stimuli, is rather damage resistant, and remains functional even if exposed to varying ambient conditions. Inspired by the fascinating combination of properties of human skin, electronic skin (e-skin) has been introduced. E-skin is capable of detecting external stimuli and hence is in high demand for smart wearables, haptic devices, and as skin for robotic applications[1–4]. E-skin is frequently fabricated from hydrogel-based resistors because they combine sensing functionality and softness[5,6]. In addition, hydrogels can be functionalized with ionic species to create ionogels, which respond to mechanical deformation[7–13]. Although ionogels have multi-sensory capabilities[14–17], they cannot assign the measured change in resistance to the stimulus that caused this change, severely limiting the value of multi-stimuli responsiveness[18–23]. Moreover, ionogels tend to dry if exposed to ambient conditions for prolonged times and freeze if subjected to cold temperatures[24]. These shortcomings limit the applicability of ionogels as e-skins, particularly in applications that expose them to varying environmental conditions[25,26]. Nature tackles this challenge by adjusting the skin composition. For example, amphibians like the wood frog species Lithobates Sylvaticus excrete choline derivatives and glucose, which combined, form a deep eutectic solvent (DES) mixture that has a low melting point. Thereby, they prevent the freezing of cells within the frog at subzero temperatures[27]. By regulating the concentration of these molecules within their body, they further delay freezing during hibernation[28]. Inspired by the wood frog, combinations of choline derivatives and polyols, such as glycerol, are used in

natural pharmaceuticals due to their biocompatibility[29–31]. If hydrogels are swollen with a large quantity of DES, they transform into organogels[32–34], which dry more slowly[35], freeze at lower temperatures[36], and in many cases self-heal[13,37]. These features are especially attractive for e-skin applications. However, these organogels have not been used as multi-stimuli responsive sensors. Moreover, organogels are most commonly cast due to their excessive softness. Proof-of-concept to 3D print organogels into 2D structures has been demonstrated[38–41]. Yet, these materials could only be formulated as thin films. The limited processability of these materials hampers their application, especially in smart wearables where more involved 3D geometries are often desirable[42,43].

Here, we introduce granular organogels that, due to their ionic conductivity, can selectively detect changes in temperature, strain, and humidity. To unambiguously assign the detected change in resistance to the stimulus that caused this change, we use machine learning. To enable the processing of organogels into intricate 3D structures through direct ink writing (DIW), we formulate them as organomicrogels. We transform granular organomicrogels into load-bearing macroscopic granular materials by connecting them with a secondary polymer network to form double network granular organogels (DNGOGs). Our DNGOGs can be extruded into 3D shapes, in contrast to conventional organogels, whose processing is typically limited to casting[38–41]. To demonstrate the potential of this material, we direct ink write customizable multi-sensory wearable skin that can be attached to a human finger, as shown in Fig. 1. The inclusion of the DES

[1]Soft Materials Laboratory, Institute of Materials (SMaL), École Polytechnique Fédérale de Lausanne, Lausanne, Switzerland. [2]Swiss National Center for Competence in Research (NCCR) Bio-inspired Materials, University of Fribourg, Fribourg, Switzerland. [3]CREATE Lab, Institute of Mechanical Engineering, École Polytechnique Fédérale de Lausanne, Lausanne, Switzerland. ✉e-mail: esther.amstad@epfl.ch

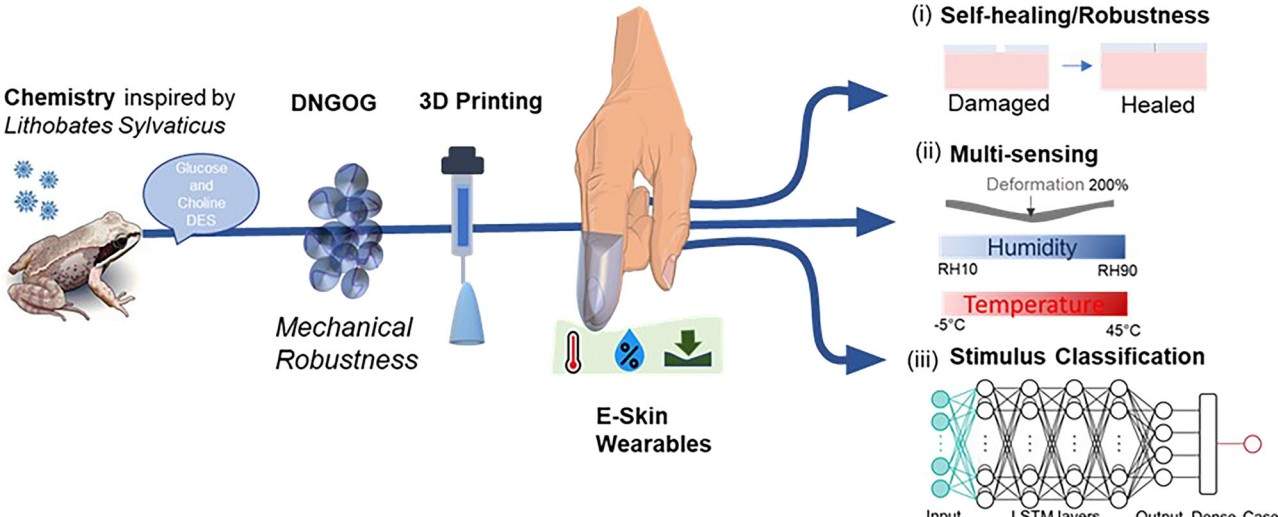

**Fig. 1 | Schematic representation of the self-healing multimodal e-skin.** The artificial e-skin is inspired by the DES chemistry of Lithobates Sylvaticus. The granular nature of the system enables 3D printing of hollow concave wearables that: (i) self-heal, (ii) can detect mechanical deformation, changes in temperature, and humidity, such that it can be used as e-skin with multimodal perception. (iii) Data-driven machine learning enables classification of the applied stimulus. The frog image is reproduced under licence from SeagerDesign-stock.adobe.com. The hand image was generated using the prompt "Human hand touching surface; Index Finger Extended" by firefly.adobe.com.

introduces self-healing properties to the DNGOG and reduces its freezing point such that this material can be used over a wide temperature range, as shown in Fig. 1i. The ionic conductivity of the organogels makes it sensitive to temperature, strain and humidity (Fig. 1ii). To classify the different stimuli, we exploit data-driven machine learning, as shown in Fig. 1iii. This combination of features is ideal for e-skin application, such that we foresee this material to open up exciting avenues in the fabrication of multi-responsive self-healing e-skin that is mechanically robust and tolerant to temperature fluctuations.

## Results and discussion
### Mechanical properties and self-healing of granular eutectogels
To introduce a 3D printable e-skin that can simultaneously detect changes in mechanical deformation, temperature, and humidity, we fabricate microgels from water-in-oil emulsions. To render our system self-healing and resistant to drying and freezing, we dissolve 2-acrylamido-2-methyl propane sulfonic acid (AMPS) in a binary solvent mixture composed of a 1:1 volume ratio of water and glycerol. The monomer containing a water/glycerol mixture is blended with mineral oil containing a surfactant and emulsified through vortexing. The resulting drops are converted into microgels by exposing them to UV light to initiate the free radical polymerization of the reagents contained in them. The microgels are transferred into an aqueous solution, and their dimensions are quantified using optical microscopy. Microgels dispersed in the binary solvent mixture are spherical with an average diameter of 21 μm and a standard deviation of 5 μm, as shown in Supplementary Fig. 1. When incubated in an aqueous solution overnight, the diameter of the microgels increases to 102 μm with a standard deviation of 12 μm, as shown in Supplementary Fig. 1.

To impart load-bearing properties and ionic conductivity to the granular organogel, we soak microgels in a DES containing acrylamide (AAm) and a crosslinker. The reagent-loaded microgels are jammed through centrifugation and cast. The granular structure is rigidified by initiating the free radical polymerization of the AAm contained within the microgels through exposure to UV light, resulting in DNGOGs. To relate the content of DES to the mechanical properties of DNGOGs, we perform tensile tests on them. We keep the molar ratio of glycerol and choline chloride constant at 1:1, while we vary the total concentration of the DES in the solution. DNGOGs made from microgels that have been swollen in a solution containing 12.5 mol% DES, which corresponds to 6.25 mol%

glycerol and 6.25 mol% choline chloride, display an ultimate tensile strength of 0.45 MPa. This value increases more than twofold if we decrease the DES concentration to 10 mol%, as shown in Fig. 2b. We assign this increase in ultimate tensile strength to an increase in the concentration of the 2nd network within DNGOGs with increasing degree of microgel swelling and hence, decreasing DES content[44]. Similarly, the increase in the 2nd network concentration increases the elongation at break from 234% to 510%, when the DES concentration is decreased from 15 to 12.5 mol%.

The stiffness of DNGOGs is typically determined by that of the microgels[45–47]. We do not vary the microgel composition or concentration such that we do not expect a change in the Young's modulus of the DNGOGs. Indeed, all DNGOGs display a Young's modulus of 0.13 MPa.

The mechanical properties of bulk eutectogels depend on the molar ratio between the hydrogen bond donor and acceptor present in the DES[40]. The optimal stoichiometric molar ratio for self-healing of choline chloride and glycerol is 1:2[48,49]. In our system, the polymer-DES interactions likely compete with the DES-DES interactions, such that we anticipate the optimal stoichiometry to be different. To identify the optimum choline chloride and glycerol ratio, we vary the glycerol content in our DNGOGs while keeping the total molar DES concentration constant at 12.5% mol. Similarly, we keep the concentrations of AAm and crosslinker constant.

The glycerol content influences the degree of swelling of microgels, which determines the amount of AAm that they can take up. Because of the importance of the amount of AAm stored within microgels on the mechanical properties of the final granular material, we quantify the degree of swelling of microgels as a function of the glycerol concentration. Microgels incubated in a solution containing 2.5 mol% glycerol, corresponding to a choline chloride to glycerol ratio of 4:1, have a diameter of 100 ± 16 μm. The diameter of microgels decreases to 84 ± 18 μm if incubated in a solution containing 8.3 mol% glycerol, corresponding to a choline chloride to glycerol ratio of 1:2. We assign this decrease of microgel diameter with increasing glycerol concentration to the decrease in the solvent quality.

The decrease in the degree of swelling of microgels with increasing glycerol concentration reduces the amount of AAm that can be loaded into them. As a result of the lower density of the polyacrylamide (PAAm) within the DNGOGs, their tensile strength decreases from 0.36 to 0.14 MPa if we increase the molar concentration of glycerol from 2.5 to 8.3 mol%, as shown in Fig. 2c. The strain at break of DNGOGs remains constant at 350% for all tested glycerol concentrations, as shown in Fig. 2c. Yet, the strain at break of

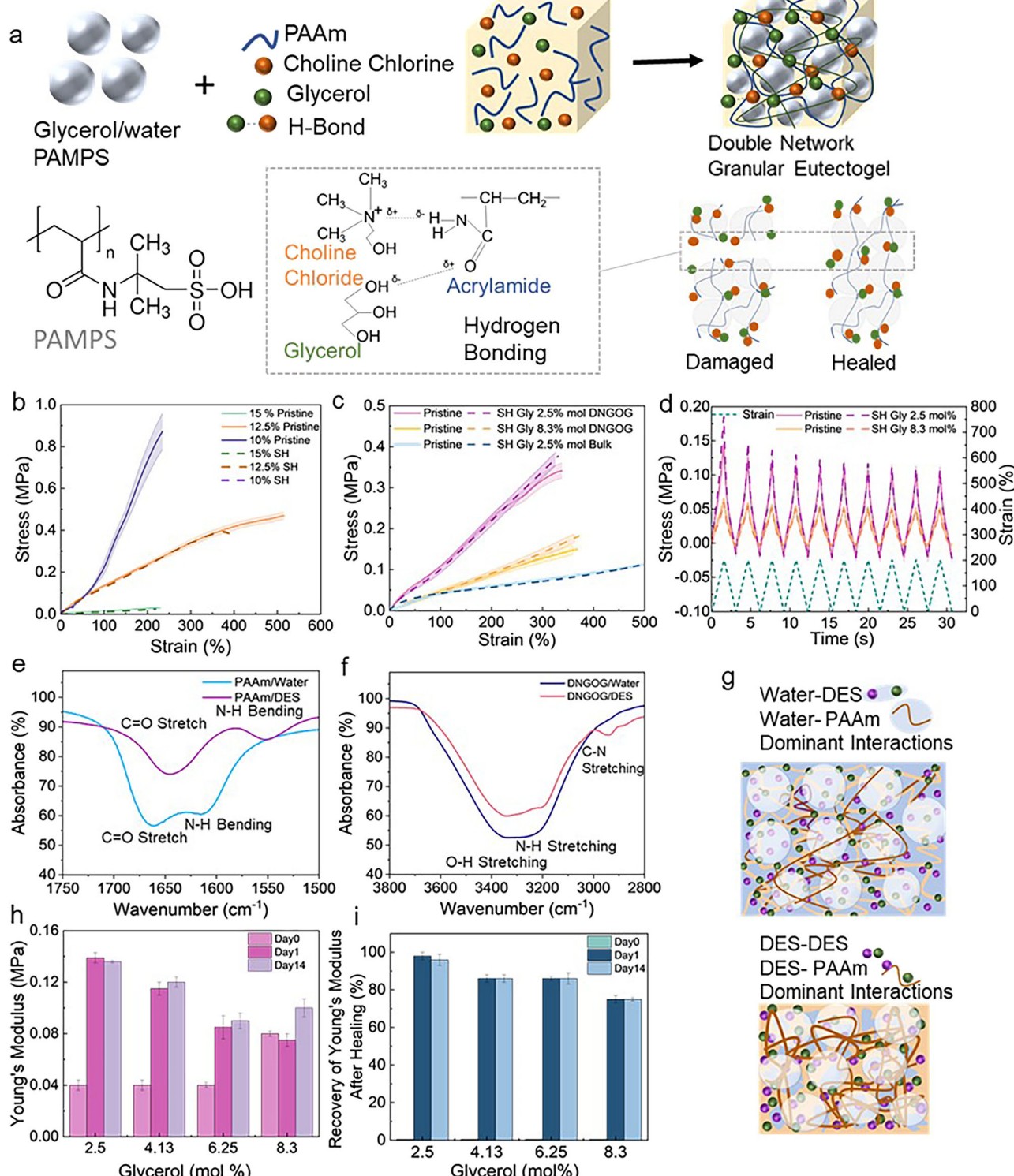

DNGOGs is significantly below that of bulk PAAm, which is 510%. We assign this result to the presence of the much stiffer and less extensible microgels in DNGOGs.

To assess the consequence of the solvent-organomicrogel interactions on the dynamic mechanical behavior of DNGOGs, we subject the material to 10 cycles of dynamic loading, where we cycle the strain between 0 and 200%. The stress at 200% strain decreases from 0.17 to 0.06 MPa if we increase the glycerol concentration from 2.5 to 8.3 mol%, as shown in Fig. 2d. We assign the measured reduction in stress at 200% strain with

increasing glycerol concentration to the reduced polymer density of the 2nd network. Overall, lower glycerol concentration in the DES leads to improved strength and Young's modulus in the DNGH.

Polymers swollen with choline chloride and glycerol tend to self-heal because strong intermolecular bonds form between the ammonium group of the choline chloride and the hydroxyl group of the glycerol[50]. To test if strong intermolecular bonds also form between the DES and the polymer network of our DNGOG, we cut 2 mm wide strips in half and put them in contact for 10 s. The strain at break of self-healed DNGOGs encompassing

**Fig. 2 | Mechanical properties of DNGOGs and their self-healing efficiency.**
**a** Components of the jammed PAMPS microgels swollen in DES made of glycerol and choline chloride. **b** Influence of the composition of the 2$^{nd}$ network on the stress-strain curve of DNGOGs subjected to tensile tests. Microgels are soaked in an aqueous solution containing 15 mol% (green), 12.5 mol% (orange), 10 mol% (purple) DES with a 1:1 molar ratio of glycerol and choline chloride, tested 1 day after production. These concentrations correspond to 7.5 mol%, 6.25 mol%, and 5 mol% glycerol in the AAm solution. **c** Tensile test of double network bulk PAMPS/PAAm functionalized with DES containing 2.5 mol% Glycerol (blue) and DNGOGs soaked with DES containing 2.5 mol% (yellow) and 8.3 mol% (orange) glycerol. **d** Dynamic tensile tests of DNGOGs soaked with DES containing 2.5 mol% (yellow) and 8.3 mol% (orange) glycerol. The composition of the DNGOGs is listed in Supplementary Table 2. The continuous line symbolizes the response of the pristine sample and the dotted line that of broken samples after they have been put in contact for 10 s at

25 °C, RH = 40%. The samples were tested 1 day after production. **e** FTIR spectra of PAAm soaked in water (blue) and DES (purple). **f** FTIR spectra of DNGOG swollen in water (navy) and DES (red). The full spectra can be found in Supplementary Fig. 2. **g** Schematic of the Deep Eutectic Solvent that hydrogen bonds to the DNGOG in the hydrated (top) and dry (bottom) state. The reduction in area represents the mass loss of DNGOGs containing 6.25% mol glycerol. Influence of the glycerol content on the mechanical properties. **h** Young's moduli and **i** recovery of the Young's moduli after broken samples have been put in contact for 10 s at 25 °C, RH = 30%, as a function of the molar ratio of glycerol contained in the DES. The measurement was performed as produced (day 0, hydrated sample), after 1 and 14 days of storage at room temperature under ambient conditions. The values are derived from Supplementary Tables 3 and 4. The error bars represent the standard deviation between three measurements.

more than 12.5% DES reaches 81% of the virgin value, as shown by the dotted line in Fig. 2b. By contrast, the strain at break of samples encompassing less than 12.5 mol% DES is as low as 10 % of the initial value after healing.

Polymers swollen with choline chloride and glycerol tend to self-heal because reversible hydrogen bonds form between the ammonium of the choline chloride and the hydroxyl of the glycerol. These results suggest that the self-healing of DNGOGs encompassing more than 12.5 mol% DES relies on strong intermolecular hydrogen bonds between the DES and the polyacrylamide of the DNGOGs that form above a critical DES concentration. To probe these interactions, we perform Fourier transform infrared (FTIR) spectroscopy on single network and DNGOG samples. The carbonyl stretch vibration of the amide group of the PAAm network shifts from 1661 to 1645 cm$^{-1}$ upon addition of DES. Similarly, the bending vibration of the amino group of PAAm shifts from 1615 to 1550 cm$^{-1}$ if DES is added, as shown in Fig. 2e. The stretching vibration of the amino group of PAAm also shifts although this shift is difficult to quantify because of the overlap with the stretching variation of the hydroxyl of the glycerol contained in the DES, as shown in Fig. 2f. These peak shifts associated with the PAAm amide group suggests that hydrogen bonds form between the PAAm amide group and the DES. These hydrogen bonds likely impart the system's self-healing properties. Since self-healing is essential for e-skins, we fix the DES concentration to 12.5 mol%, corresponding to a glycerol concentration of 6.5 mol%.

A key feature of organogels is their resistance to drying, such that they maintain their mechanical properties over time even if samples are exposed to air. As the water content decreases with the storage time, the interactions between the DES and the polymer increase, as depicted schematically in Fig. 2g.

To test the influence of the amount of water contained in DNGOGs on their mechanical properties, we characterize their mechanical properties as a function of time stored at 25 °C with a relative humidity of 40%, after 1 and 14 days of storage. The Young's modulus of organogels containing 2.5 mol% glycerol is 0.04 MPa in the hydrated state. The value increases to 0.14 MPa after 1 day of storage and does not measurably change with storage time thereafter. We assign the initial increase in Young's modulus during the first day of storage to a strengthening of interactions between DES and the polymer, which might be related to partial water evaporation during this time. Remarkably, the stress-strain curves measured on samples that have been stored for 1 and 14 days are very similar, as shown in Fig. 2h and Supplementary Fig. 3. This comparison indicates good long-term stability of samples that maintain their elasticity and strength for at least 14 days if stored under ambient conditions.

If we increase the glycerol content in the DES from 2.5 to 8.3 mol%, the Young's modulus exhibits a 30% increase when the storage time is increased from 1 to 14 days. To assess if samples dry during storage, we quantify their water content by measuring the time-dependent mass loss of the samples. Irrespective of the glycerol content in the DES, the sample loses 7% of its weight during the 1$^{st}$ day of storage and 37 wt% during 7 days of storage under ambient conditions. We cannot measure significant weight losses

between 7 and 14 days of storage, as shown in Supplementary Fig. 4. These results indicate that the majority of water contained within these samples evaporates within the first 7 days of storage. To test our assumption, we quantify the mass loss of bulk samples that contain DES and 6 wt% water. These samples lose 6 wt% if stored for 7 days, confirming that the vast majority of water is evaporated during the first week of storage, as shown in Supplementary Fig. 4c. Note that despite the water loss during the first 7 days, the relative resistance-strain curve does not change during at least 14 days of storage, hinting at the good long-term stability of the piezo-resistive response, as shown in Supplementary Fig. 3c.

Glycerol has a freezing point at −38 °C such that we expect our DNGOGs to maintain their tensile properties when exposed to subzero temperatures. To test our expectation, we compare the tensile properties of samples at −20 °C and room temperature. As expected, the elongation at break, tensile strength, and Young's modulus measured at these different temperatures are very similar, as shown in Supplementary Fig. 5a.

The solvent composition changes with storage time. Hence, we anticipate the self-healing efficiency to change during storage. The reduction in the water content should enhance hydrogen bonds between PAAm and glycerol[48], increasing their self-healing efficiency. To test this expectation, we compare the Young's modulus before damage and after healing as a function of the storage time under ambient conditions. As prepared, hydrated DNGOGs do not self-heal, due to their high-water content. By contrast, DNGOGs self-heal if they have been stored for at least 1 day: The Young's modulus of samples containing 2.5 mol% glycerol is 98% of the value of the virgin material after 10 s of contact, as shown in Fig. 2i. Even at temperatures as low as −20 °C do these samples self-heal, as shown in Supplementary Fig. 5a. We assign the good and fast self-healing of the DNGOGs to the strong DES-polymer interactions that evolve after some water has evaporated. Indeed, the recovery of the Young's modulus decreases to 74% if we increase the glycerol content to 8.3 mol% because the interactions between choline chloride and acrylamide decrease. The Young's modulus of self-healed DNGOGs does not measurably change if stored for up to 14 days, independent of the glycerol concentration within the DNGOG, as shown in Fig. 2i. Similarly, the elongation at break and ultimate tensile strength do not measurably change after healing, as shown in Fig. 2c and Supplementary Fig. 3 and summarized in Supplementary Table 1. These results indicate that strong hydrogen bonds form between the PAAm contained within the DNGOGs and choline chloride and glycerol after at least 7 wt% of water has been evaporated.

## Electrical conductivity and sensing properties of granular eutectogels

DESs are known for their high ionic conductivity that originates from the formation of charge-imbalanced ion complexes between choline chloride and the unshielded proton of glycerol, a feature that can be leveraged for sensing[51]. The ionic conductivity relies on the diffusion of ions within the gel, which we expect to scale with the viscosity of the solvent and hence, the amount of water contained in it. To assess our expectation, we measure the resistivity as a function of the water content in the DNGOGs and hence

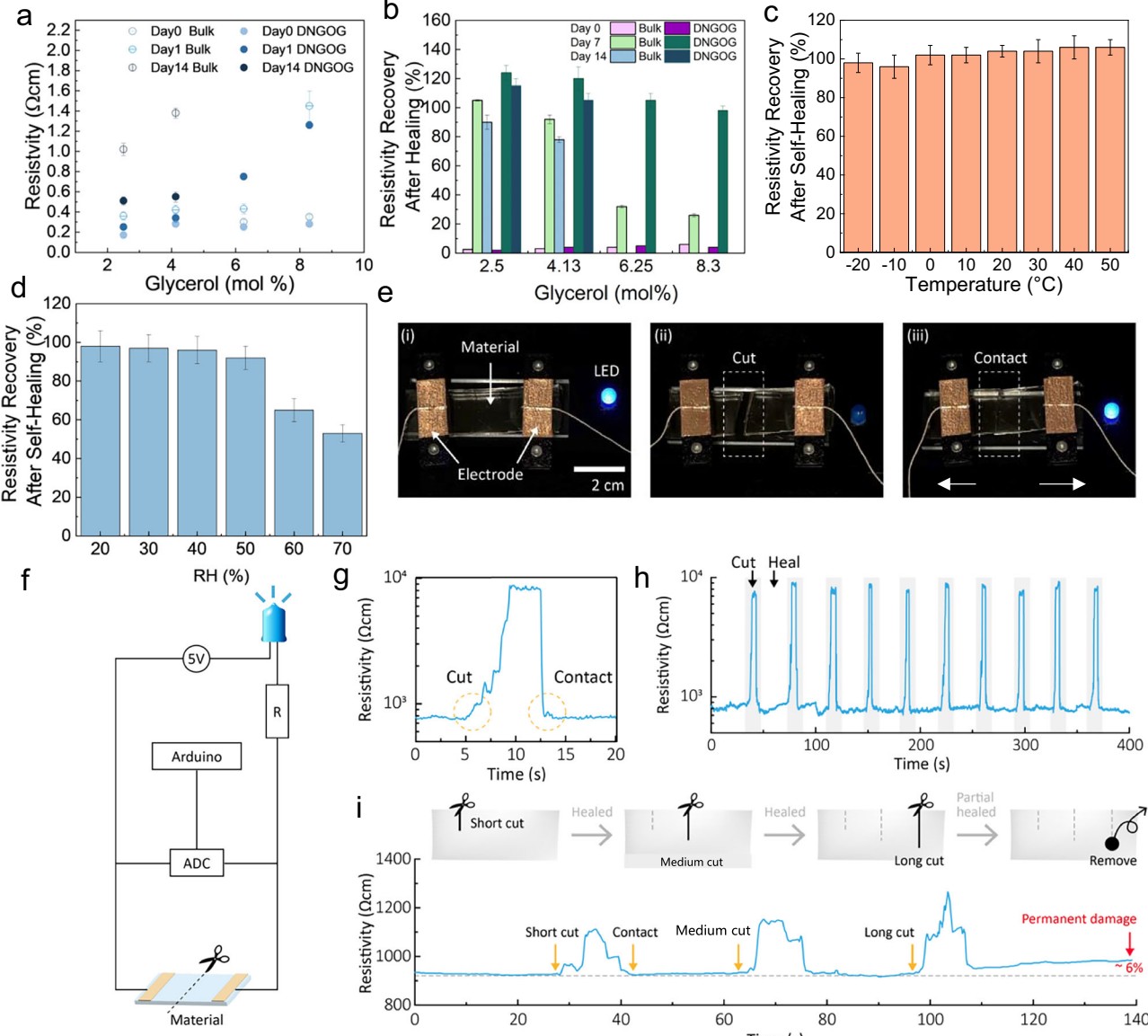

**Fig. 3 | Electrical resistivity and self-healing properties of the electrical resistivity.**
**a** Resistivity of double network bulk PAMPS/PAAm and DNGOGs with different glycerol contents in the DES as a function of the storage time. **b** Recovery of the resistivity after self-healing (contact for 10 s at 25 °C, RH = 30%) of DNGOGs and bulk samples functionalized with DES containing 2.5, 4.13, 6.25, and 8.3 mol% glycerol. The values are derived from Supplementary Table 5. The composition of the DNGOGs is listed in Supplementary Table 2. The DNGOGs with 6.25 and 8.3 mol% glycerol do not exhibit self-healing behavior after 14 days. Recovery of the resistivity for the DNGOG with 2.5 mol% glycerol after self-healing as a function of **c** temperature at RH = 30% and **d** humidity at 20 °C. **e** Photographs of the setup for cutting and healing of strips (scale bar 2 cm). **f** Schematic of the setup used to evaluate (i) multiple cycles of damage and healing and (ii) cuts with different sizes in the DNGOGs with 2.5 mol% glycerol in the DES. **g** Resistance response during the cut and self-healing of a strip in real-time. **h** The resistance response during ten cycles of severing and self-healing DNGOG films with 2.5 mol% glycerol in the DES. **i** The resistance response of DNGOGs with 2.5 mol% glycerol in the DES upon inflicting subsequent cuts with a length of 0.5, 1, and 2 cm. The samples were healed between each damage incident. The error bars represent the standard deviation between three measurements.

storage time. The resistivity of as-prepared DNGOGs is as low as 0.24 Ωcm, independent of their glycerol content, as shown in Fig. 3a. This value is similar to that of bulk eutectogels[52,53]. The resistivity of DNGOGs containing 2.5 mol% glycerol increases twofold if the DNGOG is stored for 1 day and fivefold if stored for 14 days. To assess if the increase in resistivity is caused by the decrease in water content that increases the viscosity of the DES, we quantify this parameter. Indeed, the viscosity increases with increasing glycerol content and consequently storage time, as shown in Supplementary Fig. 6. In line with these results, the viscosity of the DES increases tenfold if we deliberately increase the glycerol content from 2.5 to 8.3 mol%. Unfortunately, DNGOGs containing glycerol concentrations exceeding 6.25 mol % cannot maintain conductivity after 14 days of storage, as shown in Fig. 3a.

We attribute the loss of conductivity to a decreased mobility of the ions that is caused by the higher viscosity of these compositions. In addition, the number of charges within the DNGOGs decreases with increasing glycerol content, since the concentration of choline chloride decreases, further reducing the conductivity.

Typical ionically conductive organogels lack any microstructure. To assess the influence of DNGOG microstructure on the ionic conductivity, we quantify the resistivity of bulk organogels possessing the identical composition as DNGOGs but lacking any microstructure. Bulk samples display a threefold higher resistivity compared to DNGOGs, as shown in Fig. 3a. We assign the higher resistivity of the bulk samples to the structure of the zwitterionic PAMPS network that strongly interacts with the conductive

ions of the DES, thereby slowing down their diffusion. In bulk samples, PAMPS constitutes a continuous network. By contrast, in DNGOGs, PAMPS is only contained within the microparticles, such that their interstitial spaces are PAMPS-free. Hence, we expect the intermolecular interactions of the ions with the DNGOGs to be much weaker within their interstitial spaces, such that ions should diffuse significantly faster within these areas, leading to a lower resistivity of DNGOGs. This effect is remarkable, considering that the polymer density within bulk samples is lower than that of DNGOGs due to the jamming of the microparticles.

E-skin is prone to be damaged. To obtain a reliable sensing, the resistivity must remain unchanged after the samples self-heal. To investigate the influence of self-healing on the electrical resistivity, we sever thin strips, reattach them for 10 s, and compare the resistivity values before and after self-healing. The resistivity of DNGOGs containing 2.5 mol% glycerol attains 122% of the virgin value after samples have been healed for 1 day, as shown in Fig. 3b. The resistivity of bulk samples possessing the identical composition attains 105% of that of the virgin counterpart. The recovery of the resistivity after self-healing of DNGOGs decreases to 95% if samples contain 8.3 mol% glycerol. Remarkably, the resistivity of analog bulk samples drops to 18% after self-healing, as shown in Fig. 3b. We assign this result to the microstructure of the DNGOGs that leads to an increased number of hydrogen bonds between the DES and the gel, due to a higher concentration of DES in the interstitial spaces. The recovery of the resistivity value is independent of the storage temperature within a range of $-20\,°C$ to $50\,°C$, as shown in Fig. 3c. Even at temperatures as low as $-20\,°C$, the electrical resistivity recovers to 95% the value of the virgin counterpart, as shown in Fig. 3c. These results illustrate the broad temperature range over which our sensor can be used. By contrast, the degree of recovery of the resistivity decreases with increasing relative humidity during storage, as shown in Fig. 3d.

E-skin is often subjected to multiple damaging events. To assess the performance of DNGOGs if subjected to multiple damaging and healing events, we prepare DNGOGs containing 2.5% glycerol; this composition is chosen for its high self-healing efficiency if damaged only once, as shown in Fig. 2i. We cut the sample and heal it while monitoring its resistivity, as shown in Fig. 3e. If the resistivity is restored, the circuit is closed and the LED lights up, as shown in Fig. 3f. When the sample is cut, its resistance increases from $k\Omega$ to $M\Omega$. This resistance decreases to the initial value within 10 s, indicating that the sample is healed within this timeframe, as shown in Fig. 3g. The resistance of healed samples does not measurably change even if they have been cut and healed 10 times, as shown in Fig. 3h. To test if our DNGOGs can detect partial damage, we introduce 0.5, 1 and 2 cm cuts on the sensor. Indeed, the resistance increases with increasing length of the cut, as shown in Fig. 3i. Interestingly, if we remove a circular section from the center of the strip, we observe 6% higher values of the relative resistance, a useful feature for detecting complex defects that do not self-heal. These results suggest that our DNGOG can be used to detect complete, partial, and complex cuts on the e-skin, indicating their reliable and versatile performance as nociceptive sensors.

## Rheological characterization and 3D printing

The granular structure of DNGOGs enables their DIW into free-standing structures that can be readily customized with locally varying compositions and hence functionalities, a feature that is desirable for e-skin applications. DIW imparts stringent rheological properties to its inks. To assess the rheological properties of the granular ink, we perform amplitude sweeps on it. All tested inks are shear thinning, as shown in Fig. 4a. The inks display a low yield point, characterized as the crossover of the storage G' and loss G'' modulus, at 11% strain, as shown in Fig. 4b. Moreover, they all display a fast stress recovery, as shown in Fig. 4c. There results suggest that our inks are well-suited for DIW.

Since all tested inks display similar rheological properties, we fix the glycerol concentration to 2.5% to ensure good ionic conductivity even if stored for 7 days. To quantify the shape fidelity of our inks, we print $30 \times 30\,mm^2$ grids, as exemplified in the optical micrograph in Fig. 4di. The

spreading factor of the as-prepared ink is 31%, as shown in Fig. 4dii, a value ~threefold higher than that of analogous granular inks swollen in water[54]. We assign this difference to the presence of glycerol, which increases the viscosity of the fluid, thereby reducing the degree of jamming of the microgels. A lower degree of jamming of the microgels reduces G' and G'', as shown in Fig. 4b. The spreading factor of the DES containing granular ink decreases to 6% if stored under ambient conditions for 1 day, as shown in Fig. 4diii. We assign this strong decrease in spreading factor upon storage of the ink to the water evaporation: ~32% of the water evaporates during this time, thereby increasing the degree of jamming of the microgels.

To demonstrate the potential of our ink for DIW, we process it into a cm-sized square robot, as shown in Fig. 4e. In addition, our DNGOGs can be used to fabricate complex 3D structures, such as a cm-sized frog shown in Fig. 4f. We also 3D print personalized finger-tip sleeves as an example for wearables. Such concave geometries can be molded, but the thickness of the molded samples cannot be controlled. By contrast, 3D printing offers close control over the thickness of the finger-tip sleeves. The favorable rheological properties even enable the printing of overhanging structures, a common requirement for wearables, as exemplified on the concave wearables finger slips in Fig. 4g.

The mechanical properties of many 3D printed polymers are inferior to molded counterparts, limiting the use of many 3D printed polymers[55]. To assess if this trend also holds for the DNGOGs reported here, we perform tensile tests on 3D printed dogbones. Remarkably, the Young's modulus and elongation at break of 3D printed samples are similar to parameters measured for molded counterparts, as shown in Supplementary Fig. 7a. We assign the process-independent mechanical properties of DNGOGs to the $2^{nd}$ network that is formed after the 3D printing process has been completed, thereby firmly connecting sequentially deposited layers.

To test if the fabrication process influences the electrical properties of DNGOGs, we perform resistivity measurements on 3D printed rectangular samples. We do not see any significant difference in the electric resistivity for 3D printed and molded samples, as shown in Supplementary Fig. 7b. In fact, the recovery of the electrical resistivity increases by 15% if 3D printed samples are cut and self-healed, as shown in Supplementary Fig. 7b. We associate the higher recovery of the electrical resistivity upon self-healing of 3D printed samples compared to molded ones to the better control over their thickness that facilitates the reattachment of severed sides and thus, the formation of new choline chloride—glycerol complexes.

## Multi-sensing properties of DNGOGs

Multi-modality is key for the use of materials as e-skin sensors. We expect the resistivity of DNGOGs to change in response to humidity, temperature, and strain, such that it has the potential to be used as multimodal sensor. We first explore the resistivity response of DNGOGs to each of these stimuli individually. We subsequently leverage machine learning to distinguish and classify different stimuli.

Temperature receptors in human skin aid in the detection of contact and object classification. Hence, they are crucial for the proper functioning of e-skins. The resistivity of ion-containing organogels is caused by the ion diffusion, which is temperature-dependent. We quantify the temperature-dependence of the resistivity of DNGOGs containing 2.5 mol % glycerol, since this composition maintains electrical conductivity after drying. The relative resistance increases with decreasing temperature, shown in Fig. 5a. This temperature-dependent resistance increase can be described with the Vogel-Fulcher-Tammann (VFT) equation, suggesting that the conductivity is limited by ion diffusion. As a result of the fast decrease in mobility of the ions with decreasing temperature, the strain-dependent resistivity exponentially increases with decreasing temperature, as summarized in Supplementary Fig. 8a. By analogy, the temperature response increases with increasing humidity, as shown in Supplementary Fig. 8b. We assign this result to an increased ion mobility, caused by the higher amount of water contained in the system. Note that the change in relative resistance is independent of the DNGOG composition, as shown in Fig. 5a.

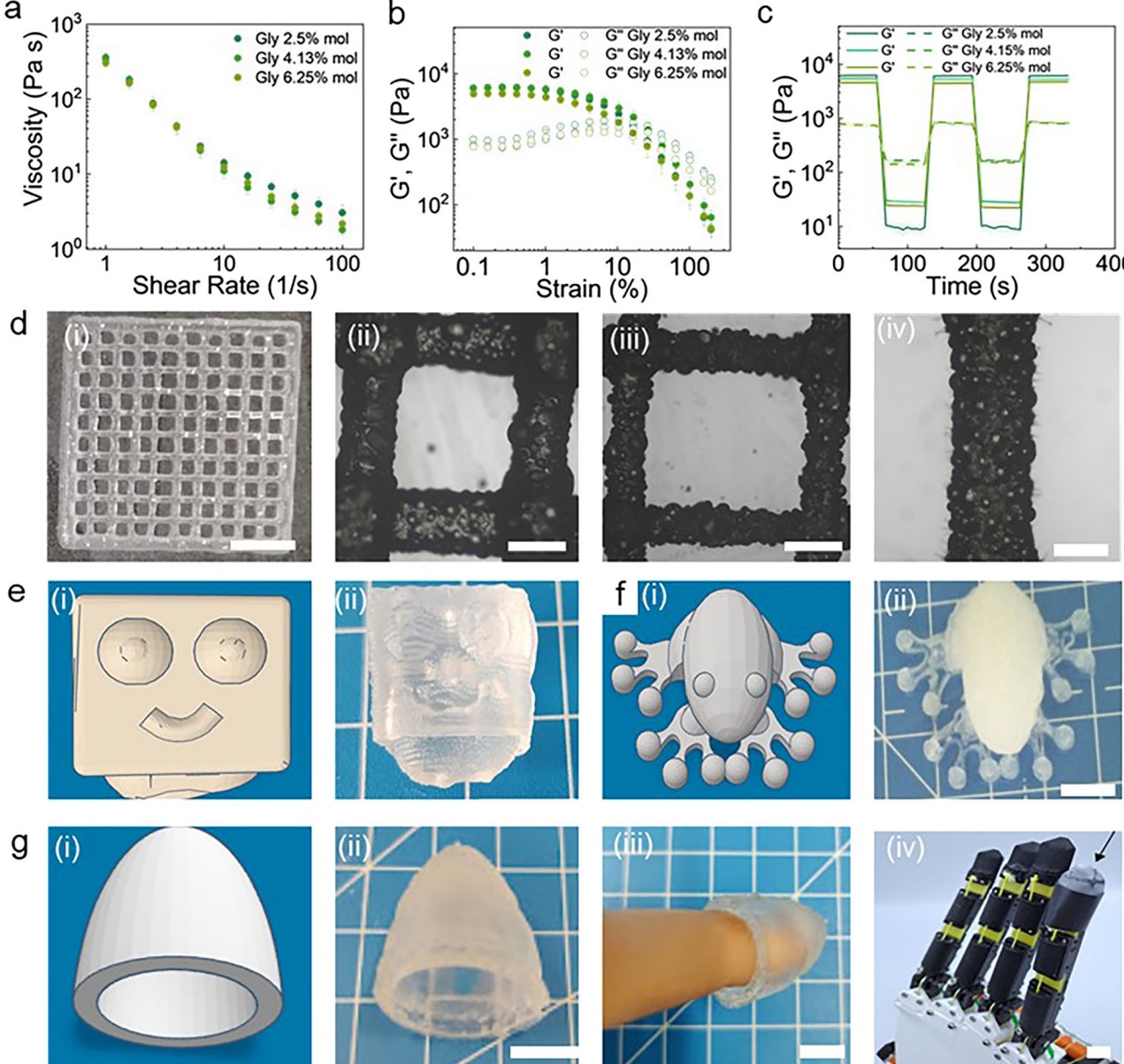

**Fig. 4 | 3D printing of DES functionalized inks.** Rheological characterization of granular inks containing 2.5, 4.13, and 6.25 mol% glycerol **a** shear rate and **b** amplitude sweeps, and **c** shear recovery tests of inks composed of jammed microgels as a function of the glycerol concentration contained in the DES. **d** Photographs of the grids printed from inks composed of DNGOGs containing 2.5 mol% glycerol (i) overview (scale bar 1 cm), (ii) hydrated grid, (iii) dried grid, and (iv) close-up of a single line of the dried grid (scale bars: 1 mm). **e** (i) CAD design (ii) 3D printed robot (scale bar: 1 cm). **f** (i) CAD design (ii) 3D printed frog (scale bar: 1 cm). **g** (i) CAD design and (ii) 3D printed wearable (iii) mounted around a finger (scale bar: 1 cm).

We quantify the signal drift of DNGOGs subjected to ten cyclic 10 K changes for starting temperatures of 0, 20, and 40 °C, respectively. The signal response in the temperature range of 10 °C to 30 °C follows the VFT equation, such that it can easily be calibrated. If the temperature is cycled from 0 to 10 °C, DNGOGs display a change in the relative resistance of 0.3 and a signal drift of 4%, as shown in Fig. 5b. If the same experiment is performed by cycling the temperature between 40 °C and 50 °C, the change in relative resistance is much smaller, 0.06, while the signal drift does not change. Remarkably, the sensor response time to changes in temperature is 20 ms, as exemplified in Supplementary Movie 1, and can detect temperature variations as small as 0.02 °C. Note that the detection limit depends on the sampling rate, which we set to 100 Hz.

The slope of the temperature-dependent resistivity decreases with increasing humidity, as shown in Supplementary Fig. 8b. This result suggests that the resistivity is humidity-dependent. Therefore, we

investigate the suitability of our DNGOGs as humidity sensors. The relative resistance of as-prepared samples that contain 32% water decreases from 0 to −0.1 if the relative humidity increases from 20% to 80%. The relative resistance decreases much more, to −0.9 if the samples have been stored for 14 days such that they do not contain significant amounts of water, as shown in Fig. 5c. The sensitivity of DNGOGs to changes in the humidity is twofold compared to that of bulk counterparts possessing the same composition, as shown in Fig. 5c. We attribute this difference to the structure of the 1st charged network. In bulk samples, this network is continuous, such that water strongly interacts with the organogel. By contrast, in DNGOGs, the 1st network is discontinuous such that water situated within the interstitial space can more easily evaporate, resulting in a faster uptake and release of water, and hence, a higher humidity sensitivity of DNGOGs compared to bulk counterparts, as shown in Supplementary Fig. 4b.

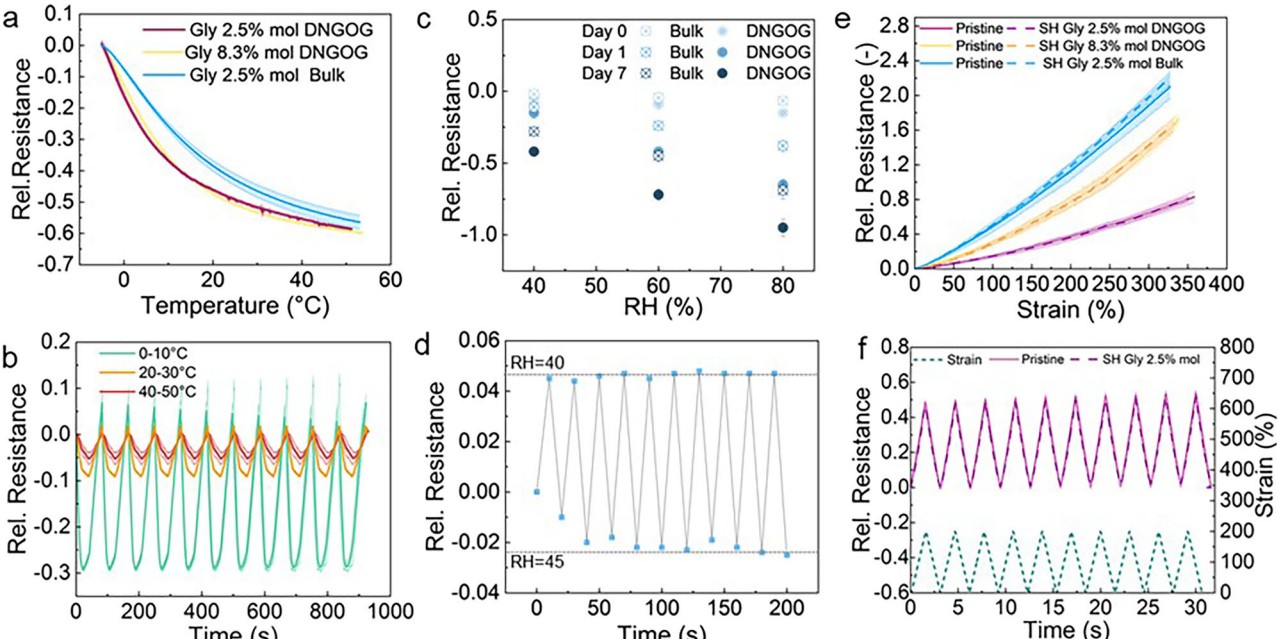

**Fig. 5 | Multi-sensing properties of DNGOGs. a** Relative resistance-temperature response of DNGOGs containing 2.5 (yellow) and 8.3 mol% (orange) glycerol measured at RH = 40%. The continuous line symbolizes the response of the pristine sample and the dotted line that of self-healed samples (contact for 10 s at 25 °C). The samples were tested 1 day after production. **b** Relative resistance response of DNGOGs containing 2.5 mol% glycerol subjected to repeated temperature changes of 10 °C starting from 0 °C (green), 20 °C (orange), and 40 °C (red). **c** Influence of the humidity on the relative resistance as a function of the storage time for DNGOGs containing 2.5 mol% glycerol. **d** Change of relative resistance if subjected to humidity cycles between 40 and 45% for DNGOGs containing 2.5 mol% glycerol. Relative resistance-strain response during **e** tensile testing to the point of fracture and **f** 10 cycles of dynamic tensile tests where the strain is cycled between 0 and 200%. The DNGOGs contain 2.5 (purple) and 8.3 mol% (orange) glycerol. The continuous line symbolizes the response of the pristine sample and the dotted line that of the self-healed sample (contact for 10 s at 25 °C, RH = 30%). The samples were tested 1 day after production, and three independent samples were tested for all assays. The error bars represent the standard deviation between three measurements.

To use DNGOGs as humidity sensors, they must repetitively change their resistivity if subjected to cyclic humidity changes. To quantify this parameter, we subject DNGOGs with 2.5 mol% glycerol to cyclic humidity changes between 40% and 45%. The DNGOGs display a drift in the resistivity of 50% for the first two cycles. Yet, the signal remains constant between the 3ʳᵈ and 10ᵗʰ cycle, as shown in Fig. 5d. This drift is not present for hydrated samples, suggesting that it originates from the drying of these samples. These results indicate that these sensors can reliably measure dynamic changes in humidity.

We expect the mobility of ions to depend on the applied strain, which would render DNGOGs piezoresistive. To test this expectation, we quantify the strain-dependent resistivity of DNGOGs containing 2.5 mol% glycerol. The relative resistance increases from 0 to 0.7 if the strain is increased to the point of fracture, exhibiting a gauge factor of 0.22 at a strain of 100%, as shown in Fig. 5e. Bulk samples exhibit a larger gauge factor of 0.45. We associate the larger gauge factor of bulk samples with a slower ion diffusion in samples lacking any microstructure. The strain-induced increase in resistance is much more pronounced if DNGOGs contain 8.3 mol% glycerol. The resistance of these samples increases by 40% if they are strained to 200%, as shown in Fig. 4e. We assign the higher strain-dependence of the resistance of DNGOGs containing more glycerol to the higher viscosity of this solution that slows down the conducting ions. Ions must diffuse over larger distances if samples are strained. The slower ion diffusion renders the resistance more sensitive to changes in the ion diffusion length, resulting in a higher piezoresistive effect. In line with this argumentation, the bulk composition exhibits a 60% higher resistance change than the corresponding DNGOG, as shown in Fig. 5e. Nevertheless, the DNGOG sensor can detect tensile strain changes as small as 0.01% if we set the sampling rate to 100 Hz. We use 2 cm long samples such that this change in strain is equivalent to a deformation of 0.05 mm.

Importantly, we do not observe any signal drift within the tested 10 cycles, as shown in Fig. 5f.

The piezoresistive response and self-healing behavior of our DNGOGs does not measurably change if they are cooled to −20 °C, as shown in Supplementary Fig. 5b. Yet, the signal noise increases at −20 °C, indicating that the electrical conductivity at this temperature is below that at room temperature, as shown in Supplementary Fig. 5b. Indeed, the measured decrease in electrical resistivity follows the trend predicted by the VFT equation. When we decrease the temperature from 25 °C to −20 °C, the electrical resistivity increases from 0.2 to 8.5 Ω cm. We assign the increase in electrical resistivity to the lower mobility of the ionic charges at lower temperatures.

The resistivity of DNGOGs is temperature-dependent. Hence, we expect the strain-dependent resistance change to also depend on the temperature. Indeed, the strain-dependent resistance change decreases with increasing temperature, as shown in Supplementary Fig. 9a. Yet, if the resistivity in the strained state is normalized by that at 0% strain, the piezoresistive response is independent of the temperature if operated at 0 °C < T < 40 °C, as shown in Supplementary Fig. 9b. Hence, this material can readily be used as a reliable strain sensor at temperatures varying from 0 to 40 °C.

**Stimulus classification using machine learning**

While 2D films are convenient for material characterization, wearable devices often have more complex shapes. For that purpose, we compare the sensor response of the 3D printed finger sleeve to that of the 2D film. The strain-resistance response of the concave finger-tip sleeve is less sensitive than that of its flat counterparts, as shown in Fig. 6a, b. We attribute this reduction in sensitivity to the simultaneous compression of the innermost parts of the finger sleeve and tension of the outermost parts of the finger.

This heterogeneous stress distribution within the wearable is in stark contrast to the homogeneous stress distribution within a thin 2D film and leads to lower sensitivity. This heterogeneous stress distribution makes a stimulus classification more difficult.

Our DNGOGs are responsive to temperature, humidity, and strain, such that they can be used as multimodal e-skin, as shown in Supplementary Movie 1. Yet, each of these stimuli results in a change in the resistivity of DNGOGs, which is not unique. Hence, if two of these stimuli are simultaneously present, the change in resistivity is superposed, as exemplified on a molded 2D strip in Fig. 6a. This superposition renders an accurate assignment of the change in resistance to the respective stimulus challenging[56,57]. For a reliable multimodal e-skin, we must classify changes in resistivity to the appropriate stimulus, even in materials with complex shapes such as concave finger-tip sleeves. To achieve this goal, we automatically gather resistivity responses to changes in strain and temperature using a robot arm equipped with a finger-tip wearing the sleeve. We monitor changes in resistivity when the temperature is changed while keeping the strain constant. Similarly, we record resistivity changes if the strain is changed, keeping the temperature constant. Finally, we simultaneously change the strain and temperature and monitor the changes in resistivity. These data are fed into a long short-term memory (LSTM) neural network, as this is well-suited for machine learning-based time-series classification. We use this network to assign the measured change in resistivity to the respective stimulus such that the sensor can distinguish between changes in temperature and strains. After the training, the output of the model exhibits the probability of the tactile event to involve a thermal or force stimulus, as shown in Fig. 6c. We employ ML to classify the sensory stimulus present during a tactile event. The stimulus classification exhibits an accuracy of 98% in the case of 2D films, as shown in Fig. 6d. After applying the ML model for the concave wearable, we can classify the stimulus with 100% accuracy, as shown in Fig. 6e. A possible reason for this higher accuracy can be linked with relaxation effects that are more prevalent in thin 2D films. Thus, in combination with ML, our e-skin can classify if changes in resistivity are caused by mechanical deformation or temperature changes, or both with high accuracy. Note that we exclude the detection of changes in humidity because sample drying is time-consuming. Hence, the inclusion of data on resistivity changes caused by alterations in humidity would have required a much lower sampling rate.

Multimodal e-skin that combines selective stimulus detection and exhibits anti-freeze/anti-drying properties has been previously reported. However, these formulations were rarely self-healing and typically could not be 3D printed. Our DNGOGs for the first time combine all these traits: the ability to sense multiple stimuli, self-heal, display anti-freeze/anti-drying properties, and 3D printability, as summarized in Fig. 6f. Furthermore, our DNGOGs display a significantly higher Young's modulus than previously reported self-healing e-skins, as summarized in Fig. 6g. We foresee these properties, combined with the longevity, to enable new applications of e-skin in wearables and robotics.

## Conclusions

We introduce an ionically conductive DNGOG that can simultaneously sense changes in strain, temperature, and humidity. We leverage the granular structure of this material to 3D print it into a load-bearing temperature, humidity, and pressure-sensitive self-healing e-skin. The material is composed of jammed PAMPS microgels that are connected through a PAAm network, resulting in DNGOGs. These organogels are swollen in a DES based on choline chloride and glycerol to impart them ionic conductivity and render them resistant to drying and freezing. We demonstrate that the ionic conductivity of DNGOGs depends on the humidity, temperature, and strain, such that they are well-suited multimodal sensors. By applying a machine learning model, we classify the sensory stimulus responsible for the change in the resistance, if multiple stimuli are present. We leverage the rheological properties of the granular paste to 3D print it into a concave wearable that we use as a multi-sensing e-skin. We foresee the

combination of multi-sensing, longevity, and processability of our DNGOGs to enhance the applicability of e-skin in soft robotics, prosthetics, and wearable medical devices.

## Methods

### Materials
AMPS (Sigma-Aldrich, USA), N, N-methylene bisacrylamide (MBA) (Carl Roth, Germany), 2-hydroxy-2-methylpropriophenone, light mineral oil (Sigma-Aldrich, USA), Span80 (TCI Chemicals, Japan), Glycerol (Sigma-Aldrich, USA), Choline Chloride (Sigma-Aldrich, USA) were used as received.

### Microgel preparation
Microgels were produced from water-in-oil emulsions. 25 wt% AMPS, 5 wt% MBA relative to the total AMPS concentration and 5 µl ml$^{-1}$ 2-Hydroxy-2-methylpropiophenone PI were dissolved in a 1:1 volume ratio water and glycerol mixture. The aqueous solution and light mineral oil containing 4 wt% Span80 used as a surfactant were combined in a 1:3 volume ratio and vortexed for 5 min to form drops. The resulting emulsion was stirred vigorously while being exposed to 60 mW cm$^{-2}$ of 390 nm UV light for 5 min to polymerize the reagents contained in the drops. Three washing steps with ethanol and three washing steps with a water/glycerol binary solvent were applied to remove the oil and surfactant. During each washing step, centrifugation at $4500 \times g$ was applied for 10 min, and the supernatant was discarded.

### Granular eutectogel sensing ink
To prepare DNGOG, microgels were soaked overnight in the AAm-containing solution. They were centrifuged at $4500 \times g$ for 10 min, and the supernatant was discarded. In addition, three filtering steps with 10 µm paper-based filters were applied to remove excess glycerol from the jammed particles. The resulting ink was cast or 3D printed before the samples were exposed to UV 366 nm, 1 mW cm$^{-1}$ for 10 min.

To evaluate the effect of the DES concentration in the DNGOG, microgels were swollen in an aqueous solution containing 25 wt% AAm, 0.3% mol MBA, 0.5 µl ml$^{-1}$ PI in relation to the AAm content and 10 mol%, 12.5 mol%, and 15 mol% DES. For this experiment, the molar ratio of glycerol and choline chloride was 1:1 for all three DES concentrations, and thus, the aforementioned compositions correspond to 5 mol%, 6.25 mol%, and 7.5 mol% glycerol contained in the AAm solution.

To evaluate the effect of the molar ratio of the glycerol and choline chloride in the DES, we soaked the microgels in an aqueous solution containing 25 wt% AAm monomer, 12.5% mol DES, and 0.3% mol MBA, 0.5 µl ml$^{-1}$ PI in relation to the AAm content. The DES was prepared by combining choline chloride and glycerol in molar ratios of 1:2, 1:1, 2:1, and 4:1, while maintaining the total concentration of DES in the AAm precursor solution constant.

### Double network bulk sensing ink
Double network bulk sensing inks were prepared by casting a solution containing 25 wt% AMPS, 5 wt% MBA relative to the total AMPS concentration, and 0.5 µl ml$^{-1}$ PI dissolved in a 1:1 volume ratio of water and glycerol mixture. The films were polymerized by exposure to UV 366 nm, 1 mW cm$^{-1}$ for 10 min. The films were swollen in a solution containing 25 wt% AAm monomer, 0.3% mol MBA, 0.5 µl ml$^{-1}$ PI in relation to the AAm and 12.5 mol% DES. The DES was prepared by combining choline chloride and glycerol in molar ratios of 1:2, 1:1, 2:1, and 4:1, while keeping the total concentration of DES in the AAm precursor solution constant. These compositions correspond to 8.3, 6.25, 4.13, and 2.5 mol% glycerol in the AAm solution. After overnight incubation in the solution, the films were exposed to UV 366 nm, 1 mW cm$^{-1}$ for 10 min.

### FTIR spectroscopy
FTIR spectra were obtained with a Spectrum 3 spectrometer (PerkinElmer, USA) in the attenuated total reflectance (ATR) mode and corrected for the

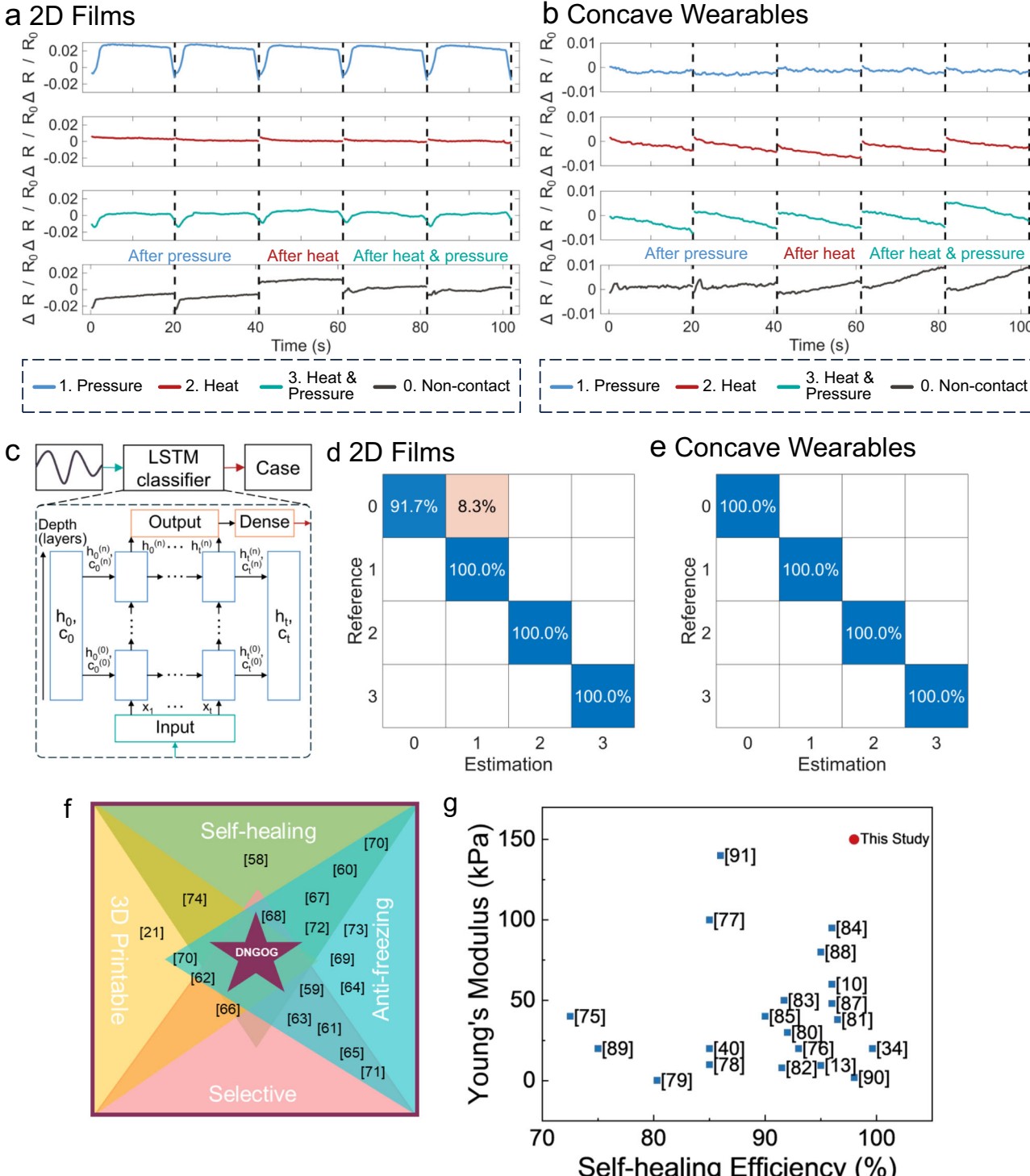

**Fig. 6 | Stimulus classification with machine learning.** Raw sensor signal after applying pressure (blue), heat (red), and combined heat and pressure (teal). **a** Thin film e-skin and **b** concave wearable e-skin. **c** Schematic of the layers of the machine learning algorithm. Confusion matrix for stimulus classification after processing the signal with the machine learning algorithm for **d** thin layer e-skin structures and **e** concave wearable e-skin. States 0, 1, 2, 3 derived from (**a, b**). **f** Mapping of studies of e-skin[21,58–74] with multiple modalities categorized by the self-healing, anti-freezing/anti-drying, 3D printability, and the selectivity to each modality. **g** Ashby plot of the Young's modulus as a function of the self-healing efficiency for e-skin based on organogel sensors[10,13,34,40,75–91].

background and $CO_2$ signals. Traces were acquired between 4000 and 800 cm$^{-1}$ at a resolution of 4 cm$^{-1}$.

**Rheological characterization**

To characterize the rheology of the ink, shear rate sweeps, strain sweeps, and shear recovery tests were performed with a DHR-3TA instrument

from Discovery (TA Instruments, New Castle, DE, USA). An 8 mm parallel plate geometry was used with a gap of 0.8 mm. The TRIOS software from TA Instruments was used for recording the data during all rheological measurements. Amplitude sweeps were performed at 1.0 rad s$^{-1}$ and subjected to strains over a range of 0.01 to 300%. Shear thinning measurements were performed in the rotational mode with a

strain rate between 1 and 100 s$^{-1}$. Shear recovery measurements were performed at 1.0 rad s$^{-1}$ and strains of 1 and 30% were applied alternatively for 60 s. The cycle was repeated two times. The viscosity of the DES solvents was measured using a Peltier concentric cylinder unit under rotational mode.

## Mechano-electrical characterization

To fabricate samples for tensile testing, the organogel was cast into dog-bone-shaped Teflon molds with a cross-section of 4.6 mm$^2$. Tensile tests were performed with a Zwick Roell Z005 universal testing machine (Zwick Roell, Ulm, Germany) with a strain rate of 200 mm/min. Ten cycles of 0–200% strain were performed with a simultaneous recording of the electrical resistance with an Arduino microcontroller equipped with an analog-to-digital converter unit. Quasi-static measurements were performed by applying five cycles of 0–50% strain with a dwell time of 30 s applied at maximum and minimum strains. The drift was calculated as the percentage difference of the relative resistance at maximum strain between the second and last cycle. The relaxation was calculated as the percentage difference of the electrical resistance at the beginning and end of the dwell time, during the quasi-static measurements. The recovery of the Young's modulus was calculated as the percentage difference between the Young's modulus before damage and after healing. The recovery of the resistivity was calculated as the percentage difference between the resistivity before damage and after healing.

## Sensing changes in ambient temperature and humidity

Changes in relative resistance with applied temperature and humidity were measured in a sealed chamber. The chamber was equipped with a temperature and humidity sensor DHT11(Adafruit Industries, New York, USA). A Polyimide-based Thermofoil heating element (Minco, Minneapolis, USA) was used for heating up the DNGOG. The heating element was connected with the temperature sensor through proportional–integral–derivative control, ensuring that the target temperature was reached and preventing overheating. Contact with ice was used for cooling below room temperature. To quantify the mass loss of the samples, the mass was measured upon production and after 1, 7, and 14 days of storage. The mass loss was defined as the percentage difference between the mass value to the original value after production.

## 3D printing

To assess the spreading factor of the ink, grids of 30 × 30 mm$^2$ were printed with infills of 10%, 15%, and 20% with a BIO X bioprinter (Göteborg, Sweden). The ink was extruded with a 22 G conical nozzle (0.41 mm diameter) through a pressure-driven piston with a speed of 10 mm s$^{-1}$. Based on the area of the cells within the grid, the spreading factor was calculated[45]. Five different cells within the grid were selected to calculate the average and standard deviation of the spreading factor. To calculate the ink spreading factor in %, the difference between the diameter of a single printed line from the theoretical value (nozzle diameter) was divided by the theoretical value of a single line (nozzle diameter).

## Machine learning model and data collection, and classification

We developed a Pytorch-based LSTM model with an input layer, followed by four layers with 64 hidden state dimensions, and an output layer with a dense layer. The LSTM model was optimized using the Adam optimizer with a learning rate of 0.001 and trained for 1024 epochs with a batch size of 256. We implemented classification by learning the patterns of the four cases (pressure, heat, pressure and heat, and non-contact). The training data was obtained via 48 repetitions for each case, and each pattern was sampled into 102 time-series data points. The trained model was tested on 12 samples for each case.

## Data availability

Data for this article, including electromechanical characterization data, rheological characterization data, and results from the machine learning

processing, are available at Zenodo: https://doi.org/10.5281/zenodo.14608429.

## Code availability

The Pytorch-based long short-term memory (LSTM) model used for the stimulus classification is available at: pytorch.org.

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

## Acknowledgements

This project was financially supported by the NCCR Bioinspired Materials (51NF40-205603) through the fellowship program Women in Science.

## Author contributions

Antonia Georgopoulou: conceptualization, validation, formal analysis, investigation, resources, data curation, writing—original draft, visualization, supervision. Sudong Lee: investigation, data curation writing—review and editing. Benhui Dai: investigation, data curation writing—review and editing. Francesca Bono: validation, investigation, writing—review and editing. Josie Hughes: validation, writing—review and editing, supervision. Esther Amstad: conceptualization, methodology, validation, formal analysis, writing—review and editing, supervision, project administration.

## Competing interests

The authors declare no competing interests.
