## [Transparent Peer Review file · Communications Materials]

3D Printing of Self-healing Longevous Multi-sensory E-Skin

Corresponding Author: Professor Esther Amstad

Version 0:

Decision Letter:

Dear Professor Amstad,

Thank you for submitting your manuscript, "3D Printing of Self-healing Longevous Multi-sensory E-Skin", to Communications Materials. It has now been seen by 4 referees, whose comments are appended below. You will see that while they find your work of potential interest, they have raised substantial concerns that must be addressed. In light of these comments, we cannot accept the manuscript for publication, but are interested in considering a revised version that addresses these serious concerns.

In particular, the Reviewers request additional experiments such as low temperature experimental data on the freezing properties, testing at a broader temperature range, and further characterisations on the temperature response time, minimum detectable temperature, force response time, minimum detectable force, and anti-freezing and water-retaining properties. They also request more details on the composition and to directly compare with state-of-the-art self-healing electronic skin.

We hope you will find the referees' comments useful as you decide how to proceed. Should further experimental data or analysis allow you to address these criticisms, we would be happy to look at a substantially revised manuscript. However, please bear in mind that we will be reluctant to approach the referees again in the absence of major revisions. If the revision process takes significantly longer than three months, we will be happy to reconsider your paper at a later date, as long as nothing similar has been accepted for publication at Communications Materials or published elsewhere in the meantime.

When submitting your revised manuscript, please include the following:

-A response letter with a point-by-point reply to each of the referee comments and a description of changes made. Please include the complete referee report in the response letter. Please note that the response letter must be separate to the cover letter to the editors.

-A marked-up version of the manuscript with all changes to the text in a different colored font. Please do not include tracked changes or comments. Please select the file type 'Revised Manuscript - Marked Up' when uploading the manuscript file to our online system.

-A clean version of the manuscript. Please select the file type 'Article File'.

-An updated <https://www.nature.com/documents/nr-editorial-policy-checklist.zip> Editorial Policy checklist, uploaded as a 'Related Manuscript File' type. This checklist is to ensure your paper complies with all relevant editorial policies. If needed, please revise your manuscript in response to these points. Please note that this form is a dynamic 'smart pdf' and must therefore be downloaded and completed in Adobe Reader. Clicking this link will download a zip file containing the pdf.

Please use the following link to submit your revised manuscript files:

Link Redacted

Please do not hesitate to contact me if you have any questions or would like to discuss the required revisions further. Thank you for the opportunity to review your work.

Best regards,

Dr Jet-Sing Lee
Senior Editor
Communications Materials
orcid.org/0000-0002-6740-8700

Reviewers' comments:

Reviewer #1 (Remarks to the Author):

The study presents a novel approach to fabricating multi-sensory e-skin using 3D printing technology, showcasing properties such as self-healing, anti-freezing, and anti-drying. However, several issues were identified during the review process:

1. While the manuscript mentions anti-freezing properties, there is no quantitative experimental data provided to support this claim. Anti-freezing is a crucial characteristic, and the manuscript would benefit from including data on the material's performance at low temperatures, demonstrating its anti-freezing capability.
2. The authors mention the material's good water retention and anti-freezing properties but report a temperature response range of only -5°C to 45°C. Given the material's water retention and anti-freezing capabilities, the temperature response range should logically cover a broader range.
3. As a sensor, the manuscript lacks detailed characterization of fundamental sensor performance, such as temperature response time, minimum detectable temperature, force response time, and minimum detectable force.
4. The manuscript claims that the sensor can distinguish between pressure and temperature stimuli, but the principle behind this capability is not well explained. If the applied pressure and temperature are not constant, can the sensor still differentiate between the two stimuli?
5. In Section 2.5, the authors mention that the sensor can detect temperature, humidity, and pressure stimuli. However, in the later demonstrations, humidity sensing is not included.
6. Some recent works related to multifunctional sensors are suggested (Carbohydrate Polymers 352 (2025) 123220; Chemical Engineering Journal 500 (2024) 156800; Chemical Engineering Journal 503 (2025) 158359; Chemical Engineering Journal 424 (2021) 130418; Nano Energy 96 (2022) 107077; ACS Appl. Mater. Interfaces 2022, 14, 30268–30278 ;ACS Appl. Mater. Interfaces 2022, 14, 43833–43843).

Reviewer #2 (Remarks to the Author):

This manuscript demonstrated a temperature, humidity and pressure sensitive self-healing e-skin based on an ionically conductive double network granular organogel which could be 3D printed into a concave wearable. The manuscript can be accepted for publication in Communications Materials. However, there are some concerns need to be addressed.

1. The illustration of the double-network hydrogel in Figure 2a is not clear enough. Please accurately indicate the composition of each network.
2. Is the eutectic composition mentioned in the manuscript a combination of choline chloride, glycerol, and acrylamide? Please provide the information of the eutectic solvents when the molar ratio of glycerol is varied.
3. The authors repeatedly referred to the double-network gel as either a hydrogel or an organogel. Please unify the terminology.
4. Please define the terms related to repair efficiency mentioned in the manuscript. For examples: recovery of Young's Modulus after healing (Figure 2i), resistivity recovery after self-healing (Figure 3c, Figure 3d.)
5. It seems that some of the data in the figure are not displayed in Figure 2i and Figure 3b). Please check it.
6. As reported in the literature, the role of water in deep eutectic solvent (DES) systems varies with its content (Effect of water on the structure and dynamics of choline chloride/glycerol eutectic systems, Journal of Molecular Liquids, 2021, 342, 117463). To further discuss the interactions between water and the DES in this gel, the water content should be considered (Figure 2g).
7. The manuscript mentioned that "These results suggest that the water content within DNGOGs does not measurably influence their Young's modulus" (Line 199 Page 10). Please try to explain this phenomenon.
8. The manuscript mentioned that "These results indicate that the formation of hydrogen bonds between the PAAm contained within the DNGOGs and choline chloride and glycerol and is completed within 1 day" (Line 211 Page 19). Please provide proof like FTIR spectra.
9. As shown in Figure 5e, the bulk gel exhibited more pronounced changes in resistivity in response to strain stimuli. Why not use the bulk gel as a sensor?
10. "Wavenumber (cm-1)" in Figure S2 should be corrected.

Reviewer #3 (Remarks to the Author):

The present manuscript describes the design and preparation of a 3D printable, self-healing, conductive organogel as temperature, strain, and humidity sensors for e-skin application. Overall, this work is valuable for the design and application of multifunctional conductive gels, and the experimental design is reasonable. This manuscript should be considered for its publication in Communications Materials after the following points listed below are addressed.

1. The novelty and design ideas of the present work were not well explained in the "Introduction" part. Furthermore, some important literatures about multifunctional conductive gels are recommended to be cited and reviewed in the introduction part and the results and discussion section, for example, 1) Journal of Materials Science & Technology 181 (2024) 91-103; 2) Collagen & Leather, 5(1), 17 (2023) <https://doi.org/10.1186/s42825-023-00123-9>; 3) Journal of Materials Chemistry B, 2024, 12, 6940-6958.
2. A table is recommended to show the detailed components of the prepared gels in this work.
3. The anti-freezing and water-retaining properties of the present gel should be well characterized.
4. The English of the whole manuscript should be greatly improved and carefully polished to correct some grammatical mistakes, spelling errors and minor errors.

Reviewer #4 (Remarks to the Author):

This work presents a novel double-network granular organogel (DNGOG) inspired by the deep eutectic solvent (DES) chemistry of Lithobates Sylvaticus, which integrates multi-sensory capabilities (strain, temperature, humidity), self-healing properties, and 3D printability for e-skin applications. The combination of material innovation, machine learning-driven stimulus classification, and advanced manufacturing techniques is commendable. The study addresses critical limitations of existing ionogels, such as environmental instability and lack of selective stimulus detection, and demonstrates potential for wearable and robotic applications. However, there are still some pressing issues that require major revisions to enhance the impact and reproducibility of the manuscript.

1. The manuscript lacks a direct comparison with state-of-the-art self-healing electronic skin. The Ashby plot in Figure 6g is inadequate, should address recent work and provide a quantitative benchmark.
2. The claim of "anti-drying" properties appears contradictory to the 37% mass loss attributed to water evaporation after 7 days (Figure S4). The authors should clarify how the residual DES composition maintains functionality despite significant water loss.
3. The higher classification accuracy for concave wearables (100%) compared to 2D films (98%) warrants a detailed explanation. Potential factors could include structural differences influencing stress distribution or variations in sensor placement during data acquisition.
4. While FTIR shifts in Figure 2e-f imply DES-polymer interactions, the manuscript lacks a mechanistic explanation of how dynamic bonds contribute to self-healing.
5. The recovery data in Figure 3 focuses solely on electrical properties, omitting critical mechanical recovery metrics (e.g., tensile strength or elasticity after healing). Quantifying mechanical self-healing efficiency after multiple damage cycles and under extreme conditions is essential to evaluate robustness for real-world applications.
6. The demonstration of 3D printed wearables (Figure 4e-g) focuses on geometric complexity but lacks functional validation. To substantiate claims of "customizable multi-sensory" capabilities, the authors should provide mechanical and electrical performance data for printed structures.
7. The introduction would benefit from a more comprehensive review of recent advancements in machine learning-driven sensing recognition (e.g., Adv. Funct. Mater., 2024, 35, 2414811; Adv. Funct. Mater., 2024, 34, 2411688; Nano Energy, 2024, 127, 109799).

Communications Materials is committed to improving transparency in authorship. As part of our efforts in this direction, we are now requesting that all authors identified as 'corresponding author' create and link their Open Researcher and Contributor Identifier (ORCID) with their account on the Manuscript Tracking System prior to acceptance. ORCID helps the scientific community achieve unambiguous attribution of all scholarly contributions. You can create and link your ORCID from the home page of the Manuscript Tracking System by clicking on 'Modify my Springer Nature account' and following the instructions in the link below. Please also inform all co-authors that they can add their ORCIDs to their accounts and that they must do so prior to acceptance.

Version 1:

Decision Letter:

Dear Professor Amstad,

Thank you for submitting your manuscript, "3D Printing of Self-healing Longevous Multi-sensory E-Skin", to Communications Materials. It has now been seen again by 4 referees, whose comments are appended below. You will see that while they Reviewers 1 to 3 think your work is acceptable, some important points are still raised by Reviewer 4. We are interested in the possibility of publishing your study in Communications Materials, but would like to consider your response to these concerns in the form of a revised manuscript before we make a decision on publication.

In particular, Reviewer 4 still has remaining questions, asking for the corresponding decoupling processing experiments and testing other basic performance parameters.

We therefore invite you to revise and resubmit your manuscript, taking into account the points raised.

When submitting your revised manuscript, please include the following:

-A response letter with a point-by-point reply to each of the referee comments and a description of changes made. Please include the complete referee report in the response letter. Please note that the response letter must be separate to the cover letter to the editors.

-A marked-up version of the manuscript with all changes to the text in a different colored font. Please do not include tracked changes or comments. Please select the file type 'Revised Manuscript - Marked Up' when uploading the manuscript file to our online system.

-A clean version of the manuscript. Please select the file type 'Article File'.

-An updated <https://www.nature.com/documents/nr-editorial-policy-checklist.zip> Editorial Policy checklist, uploaded as a 'Related Manuscript File' type. This checklist is to ensure your paper complies with all relevant editorial policies. If needed, please revise your manuscript in response to these points. Please note that this form is a dynamic 'smart pdf' and must therefore be downloaded and completed in Adobe Reader. Clicking this link will download a zip file containing the pdf.

In the event that your manuscript is accepted we will provide detailed guidance on our journal policies and formatting. You may however wish to ensure that the manuscript complies with our house style at this stage. See our style and formatting guide (<https://www.nature.com/documents/commsj-phys-style-formatting-guide-accept.pdf>) and checklist (<https://www.nature.com/documents/commsj-phys-style-formatting-checklist-article.pdf>) for reference.

Data availability statements and data citations policy: All Communications Materials manuscripts must include a section titled "Data Availability" at the end of the Methods section or main text (if no Methods). More information on this policy, and a list of examples, is available at <http://www.nature.com/authors/policies/data/data-availability-statements-data-citations.pdf>.

- Accession codes for deposited data
- Other unique identifiers (such as DOIs and hyperlinks for any other datasets)
- At a minimum, a statement confirming that all relevant data are available from the authors
- If applicable, a statement regarding data available with restrictions
- If a dataset has a Digital Object Identifier (DOI) as its unique identifier, we strongly encourage including this in the Reference list and citing the dataset in the Data Availability Statement.

DATA SOURCES: We strongly encourage authors to deposit all new data associated with the paper in a persistent repository where they can be freely and enduringly accessed. We recommend submitting the data to discipline-specific, community-recognized repositories, where possible and a list of recommended repositories is provided at <http://www.nature.com/sdata/policies/repositories>.

If a community resource is unavailable, data can be submitted to generalist repositories such as <https://figshare.com/> or <http://datadryad.org/> Dryad Digital Repository. Please provide a

unique identifier for the data (for example a DOI or a permanent URL) in the data availability statement, if possible. If the repository does not provide identifiers, we encourage authors to supply the search terms that will return the data. For data that have been obtained from publically available sources, please provide a URL and the specific data product name in the data availability statement. Data with a DOI should be further cited in the methods reference section.

Please use the following link to submit your documents:

Link Redacted

We hope to receive your revised paper within three months; please let us know if you aren't able to submit it within this time so that we can discuss how best to proceed. If we don't hear from you, and the revision process takes significantly longer, we will close your file. In this event, we will still be happy to reconsider your paper at a later date, as long as nothing similar has been accepted for publication at Communications Materials or published elsewhere in the meantime.

Please do not hesitate to contact me if you have any questions or would like to discuss these revisions further. We look forward to seeing the revised manuscript and thank you for the opportunity to review your work.

Best regards,

Dr Jet-Sing Lee
Senior Editor
Communications Materials
orcid.org/0000-0002-6740-8700

Reviewers' comments:

Reviewer #1 (Remarks to the Author):

accept.

Reviewer #2 (Remarks to the Author):

The authors have addressed all the concerns raised by the reviewers. The manuscript can be accepted without further revision.

Reviewer #3 (Remarks to the Author):

This manuscript was well revised, and should be accepted for publication.

Reviewer #4 (Remarks to the Author):

The authors have made adequate modifications according to the previous comments, but there are still some other issues needed to be solved before considering it to be accepted for publication. The details can be listed as follows:

1 In Fig. 5b, why the base line of the sensor under 0~10 is higher than other temperature range?

2 The temperature sensing performance of the sensor is conducted under 40% RH, so how about the sensing behavior of the sensor under other humidity conditions?

3 More importantly, whether the external temperature can affect the strain sensing performances for the sensor? If so, the corresponding decoupling processing is necessary.

4 Some other basic sensing performance should also be systematically studied, such as the response time, sensitivity, long-term sensing stability.

Communications Materials is committed to improving transparency in authorship. As part of our efforts in this direction, we are now requesting that all authors identified as 'corresponding author' create and link their Open Researcher and Contributor Identifier (ORCID) with their account on the Manuscript Tracking System prior to acceptance. ORCID helps the scientific community achieve unambiguous attribution of all scholarly contributions. You can create and link your ORCID from the home page of the Manuscript Tracking System by clicking on 'Modify my Springer Nature account' and following the instructions in the link below. Please also inform all co-authors that they can add their ORCIDs to their accounts and that they must do so prior to acceptance.

Version 2:

Decision Letter:

Dear Professor Amstad,

Your manuscript titled "3D Printing of Self-healing Longevous Multi-sensory E-Skin" has now been seen again by our referees, whose comments appear below. In light of their advice I am delighted to say that we are happy, in principle, to publish a suitably revised version in Communications Materials.

We therefore invite you to edit your manuscript to comply with our journal policies and formatting style in order to maximise the accessibility and therefore the impact of your work.

EDITORIAL REQUESTS

* Your manuscript should comply with our policies and format requirements, detailed in our style and formatting guide (<https://www.nature.com/documents/commsj-phys-style-formatting-guide-accept.pdf>).

* Please edit your manuscript according to the editorial requests in the attached table, and outline revisions made in the right hand column. If you have any questions or concerns about any of our requests, please do not hesitate to contact me. It is important that each request be addressed in order to avoid delays in accepting your manuscript. Please upload the completed table with your manuscript files as a Related Manuscript file.

* The editorial requests table also includes a full list of the files that must be provided upon resubmission. Please upload your files according to this table.

* An updated editorial policy checklist that verifies compliance with all required editorial policies must be completed and uploaded with the revised manuscript. All points on the policy checklist must be addressed; if needed, please revise your manuscript in response to these points. Please note that this form is a dynamic 'smart pdf' and must therefore be downloaded and completed in Adobe Reader. Clicking this link will download a zip file containing the pdf.

OPEN ACCESS

Communications Materials is a fully open access journal. Articles are made freely accessible on publication. For further information about article processing charges, open access funding, and advice and support from Nature Research, please visit <https://www.nature.com/commsmat/open-access>

Please use the following link to submit your revised files:

Link Redacted

We hope to hear from you within two weeks; please let us know if the process may take longer.

Best regards,

Dr Jet-Sing Lee
Senior Editor
Communications Materials
orcid.org/0000-0002-6740-8700

REVIEWERS' COMMENTS:

Reviewer #4 (Remarks to the Author):

The author has made adequate modifications, and I am glad to recommend it to be considered for publication in its current version.

Point-by-point answer

We thank all the reviewers for their insightful and constructive feedback. Based on these comments, we modified the manuscript and performed additional experiments. We added low temperature experimental data on the freezing properties, including temperatures as low as -20°C . Moreover, we characterized the temperature response time, minimum detectable temperature, force response time and minimum detectable force. We included measurements that compare the mechanical properties of 3D printed and moulded samples. We feel that these changes in response to the reviewers' comments significantly improved its clarity and quality. In addition, we carefully revised the manuscript to correct grammar, spelling and minor errors.

Reviewer 1

Question 1: While the manuscript mentions anti-freezing properties, there is no quantitative experimental data provided to support this claim. Anti-freezing is a crucial characteristic, and the manuscript would benefit from including data on the material's performance at low temperatures, demonstrating its anti-freezing capability.

Answer: We would like to thank the reviewer for this suggestion. Indeed, further investigation of the anti-freezing properties provides useful insights about the material properties. Thus, we included tensile characterization to compare the material properties at -20°C and 25°C . In addition, we added a comparison of the self-healing efficiency at different temperatures in a range -20°C to 50°C .

In page 10 and line 15 we added: "Glycerol has a freezing point of -38°C such that we expect our DNGOGs to maintain their tensile properties when exposed to subzero temperatures. To test our expectation, we compare the tensile properties of samples at -20°C and room temperature. As expected, the elongation at break, tensile strength and Young's modulus measured at these different temperatures are very similar, as shown in Figure S5a."

Figure S5: a) Stress-strain response and b) relative resistance-strain response of the DNGOG with 2.5 mol% Glycerol at -20°C (blue) and 25°C (red). The continuous line symbolizes the response of the pristine sample and the dotted line that of broken samples after they have been put in contact for 10 s at $\text{RH}=40\%$.

In page 20 and line 22 we added: "The piezoresistive response and self-healing behavior of our DNGOGs does not measurably change if they are cooled to -20°C , as shown in Figure S5b. Yet,

the signal noise increases at -20°C , indicating that the electrical conductivity at this temperature is below that at room temperature, as shown in Figure S5b. Indeed, the measured decrease in electrical resistivity follows the trend predicted by the Vogel-Fulcher-Tammann equation. When we decrease the temperature from 25°C to -20°C , the electrical resistivity increases from $0.2\ \Omega\text{cm}$ to $8.5\ \Omega\text{cm}$. We assign the increase in electrical resistivity to the lower mobility of the ionic charges at lower temperatures.”

We also extended the temperature range in Figure 3c to include the self-healing efficiency at -20°C :

Figure 3c: Temperature-dependent recovery of the resistivity of DNGOGs containing 2.5 mol% glycerol after self-healing at $\text{RH}=40\%$.

Question 2: The authors mention the material's good water retention and anti-freezing properties but report a temperature response range of only -5°C to 45°C . Given the material's water retention and anti-freezing capabilities, the temperature response range should logically cover a broader range.

Answer: We thank the reviewer for this comment. We would like to clarify that our material does not have good water retention properties. We obtain anti-freeze and longevity properties because we add glycerol to our system to obtain a deep eutectic solvent that forms after the majority of the water initially contained in the formulation evaporates. This is the case after day 1 of storage. We agree with the reviewer that the temperature range should be increased and for that reason we added measurements that were conducted down to temperatures of -20°C , as shown in the new Figure 3c.

Question 3: As a sensor, the manuscript lacks detailed characterization of fundamental sensor performance, such as temperature response time, minimum detectable temperature, force response time, and minimum detectable force.

Answer: We thank the reviewer for this constructive feedback. To address these good questions, we quantified these properties and report them in the revised manuscript in page 18 and line 10 as: “Remarkably, the sensor responds to changes in temperature within 20 ms, as exemplified in Movie S1 and can detect temperature variations as small as 0.02°C . Note that the detection limit depends on the sampling rate, which we set to 100 Hz.

The sensor can detect tensile strain changes as small as 0.01% if we set the sampling rate to 100 Hz. We use 2 cm long samples such that this change in strain is equivalent to a deformation of $50\ \mu\text{m}$.”

Question 4: The manuscript claims that the sensor can distinguish between pressure and temperature stimuli, but the principle behind this capability is not well explained. If the applied pressure and temperature are not constant, can the sensor still differentiate between the two stimuli?

Answer: We would like to thank the reviewer for this question. Indeed, the sensor response cannot differentiate between temperature and pressure stimuli because the change in resistivity caused by these two stimuli is superposed, as exemplified on a molded 2D strip in Figure 6a. To differentiate between these two stimuli, we use machine learning, where a computational model is established based on a series of training data to predict the correct stimulus label for resistive sensing input data. After training, the model can classify if the change in resistance is caused by temperature, strain or simultaneously by both stimuli. To clarify this important point, we modified the discussion in page 21 and line 24:

“To achieve this goal, we automatically gather resistivity responses to changes in strain and temperature using a robot arm equipped with a fingertip wearing the sleeve. We monitor changes in resistivity if the temperature is changed while keeping the strain constant. Similarly, we record resistivity changes if the strain is changed, keeping the temperature constant. Finally, we simultaneously change the strain and temperature and monitor the changes in resistivity. These data are fed into a long short-term memory (LSTM) neural network, as this is well suited for machine learning-based time series classification. We use this network to assign the measured change in resistivity to the respective stimulus such that the sensor can distinguish between changes in temperature and strains.”

Question 5: In Section 2.5, the authors mention that the sensor can detect temperature, humidity, and pressure stimuli. However, in the later demonstrations, humidity sensing is not included.

Answer: The reviewer brought up an interesting point regarding the humidity detection. Unfortunately changes in humidity can not be quickly reversed because it takes time to dry the samples. Because of the lengthy drying process, a repeated detection of humidity changes is only possible if performed over long times. To clarify this very good point, we modified the discussion in page 22 and line 14: “Note that we exclude the detection of changes in humidity because sample drying is time consuming. Hence, the inclusion of data on resistivity changes caused by alterations in humidity would have required a much lower sampling rate.”

Question 6: Some recent works related to multifunctional sensors are suggested (Carbohydrate Polymers 352 (2025) 123220; Chemical Engineering Journal 500 (2024) 156800; Chemical Engineering Journal 503 (2025) 158359; Chemical Engineering Journal 424 (2021) 130418; Nano Energy 96 (2022) 107077; ACS Appl. Mater. Interfaces 2022, 14, 30268–30278; ACS Appl. Mater. Interfaces 2022, 14, 43833–43843

Answer: We would like to thank the reviewer for the suggestions. References related to sensing hydrogel materials were added to the manuscript: Carbohydrate Polymers 352 (2025) 123220; Chemical Engineering Journal 500 (2024) 156800; Chemical Engineering Journal 424 (2021) 130418; Nano Energy 96 (2022) 107077; ACS Appl. Mater. Interfaces 2022, 14, 30268–30278; ACS Appl. Mater. Interfaces 2022, 14, 43833–43843. We could not find article 158359 in the Chemical Engineering Journal 503 (2025).

Reviewer 2

Question 1: The illustration of the double-network hydrogel in Figure 2a is not clear enough. Please accurately indicate the composition of each network.

Answer: We would like to thank the reviewer for this constructive comment. We have revised Figure 2a to include the chemical formulas of the different polymer networks present in the DNGOG.

Question 2: Is the eutectic composition mentioned in the manuscript a combination of choline chloride, glycerol, and acrylamide? Please provide the information of the eutectic solvents when the molar ratio of glycerol is varied.

Answer: We thank the reviewer for this good question. The eutectic composition involves choline chloride and glycerol. We combined this deep eutectic solvent (DES) with an aqueous solution containing acrylamide. To clarify the compositions of our different formulations, we now include a new Table S2 that lists all components with their concentrations.

Table S2. Formulation of DNGOGs with varying molar ratio between glycerol and choline chloride

	PAMPS	Acrylamide	Glycerol	Choline Chloride	MBA	PI*
	Microgels					
	mol (%)	mol (%)	mol (%)	mol (%)	mol (%)	mol (%)
Gly-2.5	0.14	10	2.5	10	0.03	0.9
Gly-4.13	0.14	10	4.13	8.3	0.03	0.9
Gly-6.25	0.14	10	6.25	6.25	0.03	0.9
Gly-8.3	0.14	10	8.3	4.2	0.03	0.9

*Photoinitiator: 2-hydroxy-2-methylpropiophenone

Question 3: The authors repeatedly referred to the double-network gel as either a hydrogel or an organogel. Please unify the terminology.

Answer: We would like to thank the reviewer for this comment. We carefully revised the manuscript to unify the terminology.

Question 4: Please define the terms related to repair efficiency mentioned in the manuscript. For examples: recovery of Young’s Modulus after healing (Figure 2i), resistivity recovery after self-healing (Figure 3c, Figure 3d.)

Answer: In line with this comment, we added in page 27 and line 4: “The recovery of the Young’s modulus is calculated as the percentage difference of the Young’s modulus before damage and after healing. The recovery of the resistivity is calculated as the percentage difference of the resistivity before damage and after healing.”

Question 5: It seems that some of the data in the figure are not displayed in Figure 2i and Figure 3b). Please check it.

Answer: We would like to thank the reviewer for this comment. The data are present in Figure 2i and 3b, but because the self-healing efficiency is very low, 0.5%, for higher glycerol concentrations, the values appear to be almost zero. To clarify this point, we modified discussion as follows: “As prepared hydrated DNGOGs do not self-heal, due to their high water content. By contrast, DNGOGs self-heal if they have been stored for at least 1 day: The Young’s modulus of samples containing 2.5 mol% glycerol is 98% of the value of the virgin material after 10 s of contact, as shown in **Figure 2i**. Even at temperatures as low as -20°C do these samples self-heal, as shown in Fig. S5a. We assign the good and fast self-healing of the DNGOGs to the strong DES-polymer interactions that evolve after some water has evaporated.”

To further clarify this point, we added Tables S4 and Table S5 that provide the values of the recovery featured in Figure 2i and Figure 3b):

Table S4. Recovery of the Young’s Modulus after self-healing by putting the two parts in contact for 10 s at 25°C, RH=40% for DNGOGs containing 2.5 mol% (Gly-2.5), 4.13 mol% (Gly-4.13), 6.25 mol% (Gly-6.25) and 8.3 mol% (Gly-8.3) glycerol.

Sample	Recover of Young’s Modulus (%)		
	Day 0	Day 1	Day 14
Gly-2.5	1.2 ± 0.02	98 ± 2	96 ± 3
Gly-4.13	0.8 ± 0.02	86 ± 3	85 ± 2
Gly-6.25	0.5 ± 0.04	86 ± 1	86 ± 3
Gly-8.3	0.5 ± 0.02	75 ± 2	75 ± 1

Table S5. Recovery of the electrical resistivity after self-healing by putting the two parts in contact for 10 s at 25°C, RH=40% for DNGOGs containing 2.5 mol% (Gly-2.5), 4.13 mol% (Gly-4.13), 6.25 mol% (Gly-6.25) and 8.3 mol% (Gly-8.3) glycerol.

Sample	Recover of Electrical Resistivity (%)		
	Day 0	Day 1	Day 14
Gly-2.5	2 ± 0.01	124 ± 5	115 ± 5

Gly-4.13	4 ± 0.02	120 ± 8	105 ± 5
Gly-6.25	5 ± 0.01	105 ± 5	-
Gly-8.3	4 ± 0.01	98 ± 3	-

Question 6: As reported in the literature, the role of water in deep eutectic solvent (DES) systems varies with its content (Effect of water on the structure and dynamics of choline chloride/glycerol eutectic systems, Journal of Molecular Liquids, 2021, 342, 117463). To further discuss the interactions between water and the DES in this gel, the water content should be considered (Figure 2g).

Answer: We thank the reviewer for raising this good point. In our system, the self-healing efficiency increases with decreasing water content. We assign this behavior to a higher number of hydrogen bonds that can form between the DES and the polymer with decreasing water content contained in it, as water competes with the hydrogen bond formation between these two systems. For this reason, we quantified the water content by measuring the sample weight as a function of storage time. To clarify this good point, we added the following figure (Figure S4d) to highlight how the DNGOG water changes with the storage time.

Figure S4d: Water content for DNGOGs as a function of the storage time and amount of glycerol contained in the DES.

Question 7: The manuscript mentioned that “These results suggest that the water content within DNGOGs does not measurably influence their Young’s modulus” (Line 199 Page 10). Please try to explain this phenomenon.

Answer: We thank the reviewer for raising this interesting question. The water content in our DNGOGs decreases from 32 wt% to 25 wt% and 2 wt% if the samples are stored for one and seven days, as shown in the new Figure S4d. We assign this behavior to evaporation of water during storage. However, this is only the case for samples containing high amount of choline chloride in the DES, the hydrogen bond acceptor in our DNGOGs. For the samples with the lowest choline chloride concentration, the Young’s modulus increases 30% between day 1 and day 14 of storage. These results suggest that when choline chloride is in excess, one day is sufficient to form hydrogen bonds that stabilize our system, but for lower choline chloride concentrations, longer storage time and therefore, higher water evaporation is necessary for the formation of the hydrogen bonds.

To clarify this point, we modified the paper in page 9 and line 23: “The Young’s modulus of organogels containing 2.5 mol% glycerol is 0.04 MPa in the hydrated state. The value increases to 0.14 MPa after one day of storage and does not measurably change with storage time thereafter. We assign the initial increase in Young’s modulus during the first day of storage to a strengthening of interactions between DES and the polymer, that might be related to a partial water evaporation during this time. Remarkably, the stress-strain curves measured on samples that have been stored for 1 and 14 days are very similar.”

Question 8: The manuscript mentioned that “These results indicate that the formation of hydrogen bonds between the PAAm contained within the DNGOGs and choline chloride and glycerol and is completed within 1 day” (Line 211 Page 19). Please provide proof like FTIR spectra.

Answer: We thank the reviewer for the constructive feedback. Our FTIR analysis reveals a shift of the carbonyl stretch vibration of the PAAm amide group from 1661 cm^{-1} to 1645 cm^{-1} and in the bending vibration of the amino group of PAAm from 1615 cm^{-1} to 1550 cm^{-1} upon addition of DES, as shown in Figure 2e. These peak shifts associated with the amide group indicate the formation of intermolecular interactions between the DES and the acrylamide of the PAAm that likely impart the system self-healing properties. We cannot measure any shifts in any other peaks. To clarify this important point, we modified the discussion as: “The carbonyl stretch vibration of the amide group of the PAAm network shifts from 1661 cm^{-1} to 1645 cm^{-1} upon addition of DES. Similarly, the bending vibration of the amino group of PAAm shifts from 1615 cm^{-1} to 1550 cm^{-1} if DES is added, as shown in Figure 2e. The stretching vibration of the amino group of PAAm also shifts although this shift is difficult to quantify because of the overlap with the stretching variation of the hydroxyl of the glycerol contained in the DES, as shown in Figure 2f. These peak shifts associated with the PAAm amide group suggests that hydrogen bonds form between the PAAm amide group and the DES. These hydrogen bonds likely impart the system self-healing properties.”

Question 9: As shown in Figure 5e, the bulk gel exhibited more pronounced changes in resistivity in response to strain stimuli. Why not use the bulk gel as a sensor?

Answer: The reviewer raises an interesting point regarding the bulk samples. Indeed, Figure 5e indicates that the sensor response is more sensitive for bulk samples. Unfortunately, the bulk samples are not self-healing whereas DNGOGs are. We assign this result to the microstructure of the DNGOGs that leads to an increased number of hydrogen bonds that form between the DES and the gel, due to a higher concentration of DES in the interstitial spaces.

In addition, the bulk samples have higher electrical resistivity compared to DNGOGs, which leads to stronger changes in resistivity if strained. The interstitial spaces of DNGOGs are PAMPS-free. We expect ions to diffuse faster within PAMPS-free areas as they are not electrostatically attracted by PAMPS, thereby imparting DNGOGs a higher ionic conductivity. To clarify this good point, we modified the discussion in page 12 and line 11:

“We assign the higher resistivity of the bulk samples to the structure of the zwitterionic PAMPS network that strongly interacts with the conductive ions of the DES, thereby slowing down their diffusion. In bulk samples, PAMPS constitutes a continuous network. By contrast, in DNGOGs, PAMPS is only contained within the microparticles such that their interstitial spaces are PAMPS-free. Hence, we expect the intermolecular interactions of the ions with the DNGOGs to be much

weaker within their interstitial spaces, such that ions should diffuse significantly faster within these areas, leading to a lower resistivity of DNGOGs.”

Question 10: “Wavenumber (cm-1)” in Figure S2 should be corrected.

Answer: The authors would like to thank the reviewer for this comment. The mistake has been corrected.

Reviewer 3

Question 1: The novelty and design ideas of the present work were not well explained in the “Introduction” part. Furthermore, some important literatures about multifunctional conductive gels are recommended to be cited and reviewed in the introduction part and the results and discussion section, for example, 1) Journal of Materials Science & Technology 181 (2024) 91-103; 2) Collagen & Leather, 5(1), 17 (2023) 3) Journal of Materials Chemistry B, 2024, 12, 6940-6958.

Answer: The authors would like to thank the reviewer for the suggestion. Indeed, the proposed references show some good examples of organogels with anti-freezing properties and were added to the manuscript. In addition, we revised the introduction to highlight the novelty of our work better.

We added in page 2 and line 9: “Although ionogels have multi-sensory capabilities,¹⁴⁻¹⁷ they cannot assign the measured change in resistance to the stimulus that caused this change, severely limiting the value of multi-stimuli responsiveness, especially when multiple stimuli are present.” And in page 3 and line 3: “Here, we introduce granular organogels that due to their ionic conductivity can selectively detect changes in temperature, strain, and humidity. To unambiguously assign the detected change in resistance to the stimulus that caused this change, we use machine learning to classify stimuli.”

We added in page 2 and line 26: “organogels are most commonly cast, due to their excessive softness. Proof-of-concept to 3D print organogels into 2D structures has been demonstrated.³⁸⁻⁴¹ Yet, these materials could only be formulated as thin films. The limited processibility of these materials hampers their application especially in smart wearables where more involved 3D geometries are often desirable.^{42,43}” and in page 3 and line 9: “Our DNGOGs can be extruded into 3D shapes, in contrast to conventional organogels whose processing is typically limited to casting.³⁸⁻⁴¹ To demonstrate the potential of this material, we direct ink write customizable multi-sensory wearable skin that can be attached to a human finger”

Question 2: A table is recommended to show the detailed components of the prepared gels in this work.

Answer: The authors would like to thank the reviewer for this good suggestion. We added tables S1 and S2 in the Supplementary Information containing information regarding the components of the gels.

Table S1. Formulation of DNGOGs with varying DES concentrations

	PAMPS	Acrylamide	Glycerol	Choline Chloride	MBA	PI*
--	-------	------------	----------	------------------	-----	-----

	Microgels					
	mol (%)	mol (%)	mol (%)	mol (%)	mol (%)	mol (%)
DES10	0.14	7.5	5	5	0.02	0.9
DES12.5	0.14	10	6.25	6.25	0.03	0.9
DES15	0.14	12.5	7.5	7.5	0.04	0.9

*Photoinitiator: 2-hydroxy-2-methylpropiophenone

Table S2. Formulation of DNGOGs with varying molar ration between glycerol and choline chloride

	PAMPS Microgels	Acrylamide	Glycerol	Choline Chloride	MBA	PI*
	mol (%)	mol (%)	mol (%)	mol (%)	mol (%)	mol (%)
Gly-2.5	0.14	10	2.5	10	0.03	0.9
Gly-4.13	0.14	10	4.13	8.3	0.03	0.9
Gly-6.25	0.14	10	6.25	6.25	0.03	0.9
Gly-8.3	0.14	10	8.3	4.2	0.03	0.9

*Photoinitiator: 2-hydroxy-2-methylpropiophenone

Question 3: The anti-freezing and water-retaining properties of the present gel should be well characterized.

Answer: We thank the reviewer for this good suggestion. In response to it, we characterized the mechanical properties of our material between 25°C and -20°C, as shown in the new Figure S5 and modified the discussion accordingly.

In page 10 and line 15 we added: “Glycerol has a freezing point of -38°C such that we expect our DNGOGs to maintain their tensile properties when exposed to subzero temperatures. To test our expectation, we compare the tensile properties of samples at -20°C and room temperature. As expected, the elongation at break, tensile strength and Young’s modulus measured at these different temperatures are very similar, as shown in Figure S5a.”

Figure S5: a) Stress-strain response and b) relative resistance-strain response of the DNGOG with 2.5 mol% Glycerol at -20°C (blue) and 25°C (red). The continuous line symbolizes the response of the pristine sample and the dotted line that of broken samples after they have been put in contact for 10 s at RH=40%.

In page 20 and line 22 we added: “The piezoresistive response and self-healing behavior of our DNGOGs does not measurably change if they are cooled to -20°C, as shown in Figure S5b. Yet, the signal noise increases at -20°C, indicating that the electrical conductivity at this temperature is below that at room temperature, as shown in Figure S5b. Indeed, the measured decrease in electrical resistivity follows the trend predicted by the Vogel-Fulcher-Tammann equation. When we decrease the temperature from 25°C to -20°C, the electrical resistivity increases from 0.2 Ωcm to 8.5 Ωcm. We assign the increase in electrical resistivity to the lower mobility of the ionic charges at lower temperatures”

We would like to emphasize that our material does not have good water retention properties. The anti-freeze and longevity properties are imparted to our material through the addition of glycerol that forms a deep eutectic solvent if combined with choline chloride. Moreover, glycerol, which has a freezing point of -38°C, a value much lower than that of water, partially replaces water. Thereby, it lowers the freezing point of the formulation.

Question 4: The English of the whole manuscript should be greatly improved and carefully polished to correct some grammatical mistakes, spelling errors and minor errors.

Answer: We carefully revised the manuscript and corrected any typos and grammar errors.

Reviewer 4

Question 1: The manuscript lacks a direct comparison with state-of-the-art self-healing electronic skin. The Ashby plot in Figure 6g is inadequate, should address recent work and provide a quantitative benchmark.

Answer: The authors are grateful for this suggestion. Indeed, we found interesting studies from the end of 2024 and 2025 that we did not include in the original submission. We apologize for this omission. We carefully revised the Ashby plot to include additional 12 organogel sensor studies, as shown in the new Figure 6g.

Figure: Ashby plot of the Young's modulus as a function of the self-healing efficiency for e-skin based on organogel sensors

For benchmarking, our DNGOG multi-modal e-skin has similarities with e-skin featured in other studies. We were particularly inspired by the very nice work reported in reference [70] Jung *et al.*, *Flex. Print. Electron.* **5**, 025003 (2020) and in reference [71] Lin *et al.*, *Nano-Micro Lett.* **13**, 200 (2021). These studies show electronic skin that can detect changes in temperature and mechanical deformation and are compatible with 3D printing. Nonetheless, anti-freezing properties were not exhibited and this is an essential aspect for e-skin that is used under varying environmental conditions including subzero temperatures. In addition, reference [70] does not report self-healing properties, which is another important property for e-skin that covers the external surface of devices and is likely susceptible to structural damage during use. Our DNGOG is the only multi-sensing material that combines anti-freezing, self-healing properties, 3D printability, and the ability to classify stimuli. This comparison is illustrated in Figure 6e.

Figure: Overview of e-skins that demonstrates their ability to respond to temperature and strain stimuli selectively, self-heal, display anti-freezing properties and be 3D printable

Question 2: The claim of “anti-drying” properties appears contradictory to the 37% mass loss attributed to water evaporation after 7 days (Figure S4). The authors should clarify how the residual DES composition maintains functionality despite significant water loss.

Answer: We thank the reviewer for this good question. The mass loss in our DNGOGs is due to water evaporation. While water evaporates, the DES becomes the dominant solvent, thereby rendering our material more resistant against freezing. With the term anti-drying properties, we mean that the properties of our DNGOGs do not change after prolonged exposure to ambient conditions, in contrast to conventional hydrogel materials. However, based on these comments

and some comments of other reviewers, we realized that this terminology is misleading and apologize for the lack of clarity. In response, we removed the word anti-drying and instead write that this material maintains its mechanical properties even if stored under ambient conditions for prolonged times.

In page 10, line 7 we mention: “To assess if samples dry during storage, we quantify the amount of water contained in them by measuring the time dependent mass loss of the samples. Irrespective of the glycerol content in the DES, the samples lose 7 wt% of during the 1st day of storage and 37 wt% during 7 days of storage under ambient conditions. We cannot measure significant weight losses between 7 and 14 days of storage, as shown in Figure S4. These results indicate that the majority of water contained within these samples evaporates within the first 7 days of storage.”

Question 3: The higher classification accuracy for concave wearables (100%) compared to 2D films (98%) warrants a detailed explanation. Potential factors could include structural differences influencing stress distribution or variations in sensor placement during data acquisition.

Answer: We thank the reviewer for highlighting this interesting point. The authors agree with the reviewer that the stress distribution is different between the 2D films and concave wearables. Since the machine learning is a data-driven approach, the data collected with the concave had higher quality, leading to higher classification accuracy. To clarify this very good point, we modified the discussion:

“We attribute this reduction in sensitivity to the simultaneous compression of the innermost parts of the finger sleeve and tension of the outermost parts of the finger. This heterogeneous stress distribution within the wearable is in stark contrast to the homogeneous stress distribution within thin 2D film and leads to lower sensitivity. This heterogeneous stress distribution makes a stimulus classification more difficult.”

The sensor placement during data acquisition is distinct for the two sample types: The 2D film is probed by the robot arm, as customary in data acquisition for e-skin. By contrast, the wearable sensor is positioned on the robot arm to simulate the way a wearable is used. We would like to highlight that in both cases the resulting classification has a very high accuracy.

Question 4: While FTIR shifts in Figure 2e-f imply DES-polymer interactions, the manuscript lacks a mechanistic explanation of how dynamic bonds contribute to self-healing.

Answer: We thank the reviewer for this question. Self-healing of our DNGOGs is based on strong hydrogen bonds that form between the DES solvent and the poly(acrylamide) network of our DNGOGs. These H-bonds are reversible such that they break if the system is strained beyond a certain extent. As long as the material is deformed only within the linear elastic regime, it retains its original shape upon removal of the strain. Once relaxed, H-bonds can re-form, enabling the system to self-heal. To clarify this point, we revised the text in page 7 and line 4:

“Polymers swollen with choline chloride and glycerol tend to self-heal because reversible hydrogen bonds form between the ammonium of the choline chloride and the hydroxyl of the glycerol.

These results suggest that the self-healing of DNGOGs encompassing more than 12.5 mol% DES relies on strong intermolecular hydrogen bonds between the DES and the polyacrylamide of the

DNGOGs that form above a critical DES concentration. They demonstrate the importance of the DES for the self-healing of DNGOGs.”

Question 5: The recovery data in Figure 3 focuses solely on electrical properties, omitting critical mechanical recovery metrics (e.g., tensile strength or elasticity after healing). Quantifying mechanical self-healing efficiency after multiple damage cycles and under extreme conditions is essential to evaluate robustness for real-world applications.

Answer: We thank the reviewer for raising this important point. Indeed, we characterized the mechanical properties before and after samples have been broken and self-healed, as shown in Figures 2b and 2c. However, we agree with the reviewer that a comparison would be easier if the data were plotted in the same plot. To avoid plotting the same data twice, we now include a table with a summary of the Young’s modulus, tensile strength and elasticity after healing, as Table S3. We also modify the discussion as:

“The elongation at break and ultimate tensile strength do not measurable change after healing, as shown in Figures 2c and S3 reveals and summarized in Table S3.”

To quantify the damage and healing after ten cycles, we added Figure S7a:

Figure S7a: Stress-strain response of the DNGOG with 2.5 mol% Glycerol after cutting the samples in half and self-healing after they have been put in contact for 10 s at RH=40% one time (red) and ten times (blue).

The equivalent electrical response can be seen in Figure 3d:

Figure 3h: The resistance response during ten cycles of severing and self-healing DNGOG films containing 2.5 mol% glycerol.

From the above figures, we observe that the tensile properties and the electrical resistivity recover after ten cycles of damage.

As for extreme conditions, we updated Figure 3c to include the self-healing at -20°C and we observe even at this low temperature, the mechanical properties remain within experimental errors unchanged: the Young's modulus after self-healing recovers to 96% of the initial value if kept at -20°C , as shown in Figure S5a. We modified the discussion to highlight these very good points at page 11 and line 1:

“Even at temperatures as low as -20°C , the electrical resistivity recovers to 95% the value of the virgin counterpart, as shown in Figure 3c. These results illustrate the broad temperature range over which our sensor can be used.”

Figure 3c: Recovery of the resistivity for the DNGOG with 2.5 mol% glycerol after self-healing as a function of temperature at RH=30%.

Question 6: The demonstration of 3D printed wearables (Figure 4e-g) focuses on geometric complexity but lacks functional validation. To substantiate claims of “customizable multi-sensory” capabilities, the authors should provide mechanical and electrical performance data for printed structures.

Answer: The authors agree with the reviewer that characterization of the 3D printed properties are necessary. In line with this comment, we added the following figure featuring the mechano-electrical properties of 3D printed and moulded samples.

Figure S7. a) Stress-strain response and b) relative resistance-strain response of the DNGOG with 2.5 mol% glycerol fabricated via moulding (red) and 3D printing (purple). The continuous

line symbolizes the response of the pristine sample and the dotted line that of broken samples after they have been put in contact for 10 s at RH=40%.

In page 17 and line 1, we modified the discussion as: “The mechanical properties of many 3D printed polymers are inferior to molded counterparts, limiting the use of many 3D printed polymers.⁵⁵ To assess if this trend also holds for the DNGOGs reported here, we perform tensile tests on 3D printed dogbones. Remarkably, the Young’s modulus and elongation at break of 3D printed samples are similar to parameters measured for molded counterparts, as shown in Figure S7a. We assign the process-independent mechanical properties of DNGOGs to the fact that the 2nd network is formed after the 3D printing process has been completed, thereby firmly connecting sequentially deposited layers.

To test if the fabrication process influences the electrical properties of DNGOGs, we perform resistivity measurements on 3D printed rectangular samples. We do not see any significant difference in the electric resistivity for 3D printed and molded samples, as shown in Figure S7b. In fact, the recovery of the electrical resistivity increases by 15% if 3D printed samples are cut and self-healed, as shown in Figure S7b. We associate the higher recovery of the electrical resistivity upon self-healing of 3D printed samples compared to molded ones to the better control over their thickness that facilitates the reattachment of severed sides and thus, the re-formation of ion complexes between choline chloride and the glycerol of the DES.”

Question 7: The introduction would benefit from a more comprehensive review of recent advancements in machine learning-driven sensing recognition (e.g., Adv. Funct. Mater., 2024, 35, 2414811; Adv. Funct. Mater., 2024, 34, 2411688; Nano Energy, 2024, 127, 109799).

Answer: The authors are grateful for the suggestion. The proposed references use machine learning for classifying mechanical sensing stimulus to human gestures. They are interesting examples of flexible sensing materials and we include them in the revised manuscript. However, we would like to highlight that these works are distinct from our work presented here where we do stimulus classification for achieving selective temperature and strain sensing under varying conditions.

Point-by-point answer

We thank all the reviewers for re-evaluating our revised manuscript. We are pleased that reviewers 1-3 recognise the improvement in our work and are happy with the changes we implemented. Based on the comments of reviewer 4, we modified the revision to clarify the additional good point that were raised.

Reviewer 4

Question 1: In Fig. 5b, why the base line of the sensor under 0~10 is higher than other temperature range?

Answer: The reviewer raises an interesting point regarding the profile of the relative resistance with the temperature. The relative resistance behaviour with temperature follows the Vogel-Fulcher-Tammann equation, which predicts that the viscosity of liquids decreases drastically at low temperatures. The decreased viscosity at low temperature directly correlates with slower diffusion of ions, which results in larger strain-dependent changes in relative resistance.

To clarify this very good point, we revised the text in page 18 and line 21: “The relative resistance increases with decreasing temperature, shown in Figure 5a. This temperature-dependent resistance increase can be described with the Vogel-Fulcher-Tammann (VFT) equation, suggesting that the conductivity is limited by the ion diffusion. As a result of the fast decrease in mobility of the ions with decreasing temperature, the strain-dependent resistivity exponentially increases with decreasing temperature, as summarized in Figure S8a.”

Question 2: The temperature sensing performance of the sensor is conducted under 40% RH, so how about the sensing behavior of the sensor under other humidity conditions?

Answer: We would like to thank the reviewer for this interesting question. Indeed, the temperature-dependent diffusion of ions within our DNGOGs is directly affected by the humidity. To further explore the influence of the humidity on the temperature sensing, we assessed the relative resistance-temperature response under different humidity values and compared their decay rates, as shown in Figure S8. We observe that the decay rate decreases with increasing humidity, indicating a faster diffusion of ions at high humidity.

Figure S8. a) Exponential decay fitting of the temperature-dependent relative resistance of DNGOGs containing 2.5% glycerol, measured at RH=80%. b) Relative resistance response with temperature at relative humidity 40% (blue), 60% (red) and 80% (violet).

In line with this comment, we modified the text in page 18 and line 24: “As a result of the fast decrease in mobility of the ions with decreasing temperature, the strain-dependent resistivity exponentially increases with decreasing temperature, as summarized in Figure S8a. By analogy, the temperature response increases with increasing humidity, as shown in Figure S8b. We assign this result to an increased ion mobility, caused by the higher amount of water contained in the system.”

Question 3: More importantly, whether the external temperature can affect the strain sensing performances for the sensor? If so, the corresponding decoupling processing is necessary.

Answer: The reviewer raises a good point regarding the effect of the temperature on the strain sensing. Interestingly, if we use relative resistance values, we do not observe any changes in the sensor performance when we vary the temperature, as shown in Figure S9a.

Figure S9. Influence of temperature cycles on the resistance. The temperature was cycled by 10K starting from 0°C (green), 20°C (orange) and 40°C (red). All samples contained 2.5 mol% glycerol. b) The relative resistance-strain response of DNGOGs containing 2.5 mol% glycerol subjected to dynamic alterations in strain between 0 and 100% measured at 0°C (blue), 20°C (orange) and 40°C (red).

To clarify this interesting point, we modified the text in page 22 and line 4. “The resistivity of DNGOGs is temperature dependent. Hence, we expect the strain-dependent resistance change to also depend on the temperature. Indeed, the strain-dependent resistance change decreases with increasing temperature, as shown in Figure S9a. Yet, if the resistivity in the strained state is normalized by that at 0% strain, the piezoresistive response is independent of the temperature if operated at $0^{\circ}\text{C} < T < 40^{\circ}\text{C}$, as shown in Figure S9b. Hence, this material can readily be used as a reliable strain sensor at temperatures varying from 0 to 40 °C.”

Question 4: Some other basic sensing performance should also be systematically studied, such as the response time, sensitivity, long-term sensing stability.

Answer: We are grateful for this good suggestion. To clarify these important aspects of the sensor response we modified the text in page 19 and line 10: “the sensor response time to changes in temperature is 20 ms, as exemplified in Movie S1” and “The relative resistance increases from 0 to 0.7 if the strain is increased to the point of fracture, exhibiting a gauge factor of 0.22 at a strain of 100%, as shown in **Figure 5e**. Bulk samples exhibit a larger gauge factor of 0.45. We associate

the larger gauge factor of bulk samples to a slower ion diffusion in samples lacking any microstructure.”

In page 10 and line 4, we added: “Remarkably, the stress-strain curves measured on samples that have been stored for 1 and 14 days are very similar, as shown in Figure 2h and Figure S3. This comparison indicates good long-term stability of samples that maintain their elasticity and strength for at least 14 days if stored under ambient conditions.” And “Note that despite the water loss during the first seven days, the relative resistance-strain curve does not change during at least 14 days of storage, hinting at the good long-term stability of the piezoresistive response, as shown in Figure S3c.”